# Human RIF1 and protein phosphatase 1 stimulate DNA replication origin licensing but suppress origin activation

Shin-ichiro Hiraga[1,*] iD, Tony Ly[2], Javier Garzón[1], Zuzana Hořejší[3,†], Yoshi-nobu Ohkubo[4], Akinori Endo[2,‡], Chikashi Obuse[4], Simon J Boulton[3], Angus I Lamond[2] iD & Anne D Donaldson[1,**] iD

## Abstract

The human RIF1 protein controls DNA replication, but the molecular mechanism is largely unknown. Here, we demonstrate that human RIF1 negatively regulates DNA replication by forming a complex with protein phosphatase 1 (PP1) that limits phosphorylation-mediated activation of the MCM replicative helicase. We identify specific residues on four MCM helicase subunits that show hyperphosphorylation upon RIF1 depletion, with the regulatory N-terminal domain of MCM4 being particularly strongly affected. In addition to this role in limiting origin activation, we discover an unexpected new role for human RIF1-PP1 in mediating efficient origin licensing. Specifically, during the G1 phase of the cell cycle, RIF1-PP1 protects the origin-binding ORC1 protein from untimely phosphorylation and consequent degradation by the proteasome. Depletion of RIF1 or inhibition of PP1 destabilizes ORC1, thereby reducing origin licensing. Consistent with reduced origin licensing, RIF1-depleted cells exhibit increased spacing between active origins. Human RIF1 therefore acts as a PP1-targeting subunit that regulates DNA replication positively by stimulating the origin licensing step, and then negatively by counteracting replication origin activation.

**Keywords** MCM; ORC1; origin licensing; PP1; RIF1
**Subject Categories** Cell Cycle; DNA Replication, Repair & Recombination

## Introduction

The process of DNA replication is critical during cell proliferation for daughter cells to inherit a complete set of genomic information. DNA replication is therefore tightly controlled to prevent either under- or over-replication, with the establishment of licensed DNA replication origins and their subsequent activation closely coupled with other cell cycle events.

Best understood in budding yeast, the mechanisms controlling origin establishment and activation appear essentially conserved throughout eukaryotes [1,2]. DNA replication origins are established in G1 phase through the stepwise formation of the pre-replication complex, composed of the origin recognition complex (ORC) and minichromosome maintenance (MCM) hexamers [3]. ORC binds first to potential origin sites and recruits MCM complexes, with this "origin licensing" step assisted by Cdc6 and Cdt1 [3]. Later in the cell cycle, origin licensing is prevented by cyclin-dependent kinase (CDK) activity, which in human cells inhibits the loading of MCM complex through several mechanisms [4]. One of these is cell cycle-dependent degradation of ORC1 protein: ORC1 phosphorylation by cyclin A–CDK2 during S/G2 cell cycle phases triggers its proteasomal degradation [5,6].

Origin activation during S phase requires two protein kinase activities, Dbf4-dependent kinase (DDK) and CDK [7]. The activities of DDK and CDK depend on cell cycle-controlled accumulation of their regulatory subunits (Dbf4 and S-phase cyclins, respectively), which are upregulated at the G1/S-phase transition. DDK phosphorylates MCM complex subunits to promote the formation of CMG (Cdc45-MCM-GINS) complex, the active DNA helicase at replication forks [7]. In budding yeast, phosphorylation sites within the N-terminal region of the Mcm4 subunit are the only essential targets for DDK to initiate DNA replication [8]. DDK activity is proposed to be one of the limiting factors for origin activation in budding yeast [9,10]. CDK in contrast drives recruitment of DNA polymerase ε, with the relevant targets most clearly identified in yeast where CDK-mediated phosphorylation of Sld2 and Sld3 promotes polymerase loading [7].

---

1 Institute of Medical Sciences, School of Medicine, Medical Sciences & Nutrition, University of Aberdeen, Aberdeen, UK
2 Centre for Gene Regulation & Expression, School of Life Sciences, University of Dundee, Dundee, UK
3 The Francis Crick Institute, Clare Hall Laboratories, South Mimms, UK
4 Graduate School of Life Science, Hokkaido University, Sapporo, Hokkaido, Japan
  *Corresponding author. Tel: +44 1224 437496; E-mail: s.hiraga@abdn.ac.uk
  **Corresponding author. Tel: +44 1224 437316; E-mail: a.d.donaldson@abdn.ac.uk
  †Present address: Barts Cancer Institute, Queen Mary University of London, London, UK
  ‡Present address: Department of Biological Sciences, Tokyo Institute of Technology, Yokohama, Japan

---

We and others recently discovered an additional layer of replication control in yeast, mediated by the RIF1 protein [11–13]. RIF1 was originally identified as a yeast telomeric chromatin component [14]. The discovery that RIF1 ensures late replication of telomeres in budding yeast [15,16] was accompanied by studies identifying a role for RIF1 in genomewide regulation of DNA replication in budding yeast, fission yeast, and in mammals [17–22]. In yeast cells, the function of RIF1 in replication control is mediated by protein phosphatase 1 (PP1) [11–13]. PP1 catalytic subunits have poor substrate specificity and are intrinsically promiscuous, requiring association with a targeting subunit for correct localization and substrate recognition [23,24]. Yeast RIF1 acts as a PP1-targeting subunit, interacting with PP1 and targeting it to dephosphorylate the MCM complex. Rif1-PP1 therefore counteracts DDK to maintain low phosphorylation levels of Mcm4, particularly during G1 phase [11–13]. Rif1 contains a series of PP1 interaction motifs ("[R/K]x[V/I]x[FW]" or "[S/G]IL[K/R]") that mediate interaction with PP1. Mutating these motifs to prevent targeting of PP1 activity by Rif1 leads to hyperphosphorylation of Mcm4, allowing origin activation even when DDK activity is abnormally low. The yeast Rif1-PP1 module therefore contributes to strict cell cycle control of MCM phosphorylation, acting as a repressor of initiation that ensures the MCM helicase is kept inactive until cells are ready to begin replication.

It has been unclear to what extent the function of yeast RIF1 in regulating replication is conserved in metazoans. As in yeast, mammalian RIF1 regulates DNA replication timing, with human and mouse RIF1 involved in controlling the genomewide replication program [17,19,22]. However, some effects of mammalian RIF1 do not appear to parallel those in yeast. In particular, mouse embryonic fibroblast cells lacking RIF1 exhibit severe sensitivity to replication-inhibiting drugs [25], which is not the case for yeast *rif1* mutants [18,26] and is not an expected consequence of removing a replication repressor.

Mammalian cells possess three closely related subtypes of the PP1 catalytic subunit ($\alpha$, $\beta$, and $\gamma$) encoded by separate genes [23,24]. Human RIF1 does contain PP1-binding motifs, although their position within the protein sequence differs from their arrangement in yeast RIF1 [27]. Partly due to this structural divergence, it has been unclear whether effects of mammalian RIF1 on replication are mediated through PP1 interaction. Human and *Drosophila* RIF1 have been reported to interact with PP1 proteins [28–33], and based on co-overexpression experiments, *Drosophila* RIF1 has been suggested to act with PP1 during fly development [33]. However, there has been no direct investigation either of the importance of the PP1 motifs, or PP1 interaction, in metazoan RIF1 function.

Here, we show that the human RIF1 protein can interact with PP1 through its PP1 interaction motifs, and that RIF1-PP1 interaction is important for controlling DNA replication by limiting phosphorylation of the MCM complex, paralleling mechanisms in yeast. We also discover an unexpected requirement for human RIF1-PP1 in stimulating the licensing of DNA replication origins, by ensuring the G1-specific stabilization of ORC1 protein essential for MCM loading on origins. Our results demonstrate that human RIF1-PP1 plays a dual role in replication control—having a repressive role at the step of origin activation (a function that is conserved from yeast to mammals), as well as a positive function in supporting origin licensing that may be specific to human cells.

# Results

## Human RIF1 protein physically interacts with protein phosphatase 1 via its PP1 interaction motifs

The evolutionarily conservation of PP1 interaction motifs suggests that PP1 targeting may be a core function of eukaryotic RIF1 proteins [27]. To investigate the importance of PP1 interaction for the function of mammalian RIF1 in DNA replication control, we mutated the three PP1 interaction motifs of human RIF1 by substituting critical residues with alanine (I292A, F294A, I2181A, L2182A, V2204A, and F2206A) to create a RIF1-pp1bs allele (Fig 1A). This RIF1-pp1bs allele and wild-type RIF1 were fused at their N-termini to GFP as described [34]. The constructs were integrated at the FRT site of the Flp-In T-REx 293 human cell line, creating a set of stable cell lines with either wild-type RIF1 or RIF1-pp1bs expressed under a doxycycline-inducible promoter. Both RIF1 and RIF1-pp1bs proteins were successfully induced by addition of doxycycline (DOX), and localized to the nucleus (Fig 1B). Without DOX induction, the GFP-fused proteins were not expressed (confirmed by microscopy and Western blotting: not shown).

We performed co-immunoprecipitation experiments to test whether GFP-RIF1 interacts with PP1 depending on the presence of PP1 interaction motifs. Western analysis revealed that the ectopically expressed GFP-RIF1 can physically interact with all three PP1 isoforms (Fig 1C, lane 5). In contrast, RIF1-pp1bs was unable to interact with any PP1 isoform (lane 6), demonstrating that human RIF1 indeed interacts with PP1 through its PP1 interaction motifs. Only a small fraction of PP1 co-immunoprecipitated with this ectopically expressed GFP-RIF1, as expected since numerous human PP1-interacting proteins compete for a limited pool of PP1 proteins [28,29,35]. We could not detect PP1 in pull-down experiments against endogenous RIF1. The amount of PP1 recovered as interacting with endogenous Rif1 may simply be below our detection threshold, or alternatively, it may be that PP1 interaction causes masking of the RIF1 epitope recognized by the antibody we used for pull-down. Endogenous RIF1 has however been identified in several large-scale analyses of proteins that interact with PP1 [28,29,31,32], confirming that the binding we observe (Fig 1C) reflects a physiological interaction.

## PP1 interaction is essential for human RIF1 to control phosphorylation of MCM proteins

We next tested the importance of the RIF1-PP1 interaction in controlling the phosphorylation of MCM proteins (Fig 2A). In these experiments, endogenous RIF1 was depleted by siRNA [34], and expression of either GFP, GFP-RIF1 (wild type) or GFP-RIF1-pp1bs was induced by DOX (Fig 2A (i)). Synonymous base substitutions in the ectopically expressed GFP-RIF1 constructs make them resistant to siRNA targeted against endogenous RIF1 [34].

Depletion of endogenous RIF1 significantly increased the phosphorylation of chromatin-associated MCM4, evident from its retardation on SDS–PAGE (Fig 2A (ii), lanes 2–4 and 6). Western analysis with phospho-specific antibodies revealed increased phosphorylation of MCM2 at sites Ser40 and Ser53 (Fig 2A (ii), lower two panels), residues known to be targets of DDK [36,37]. Note that

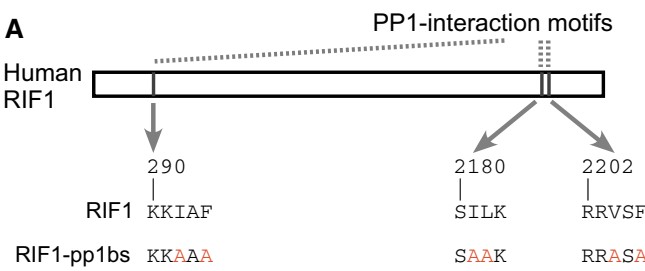

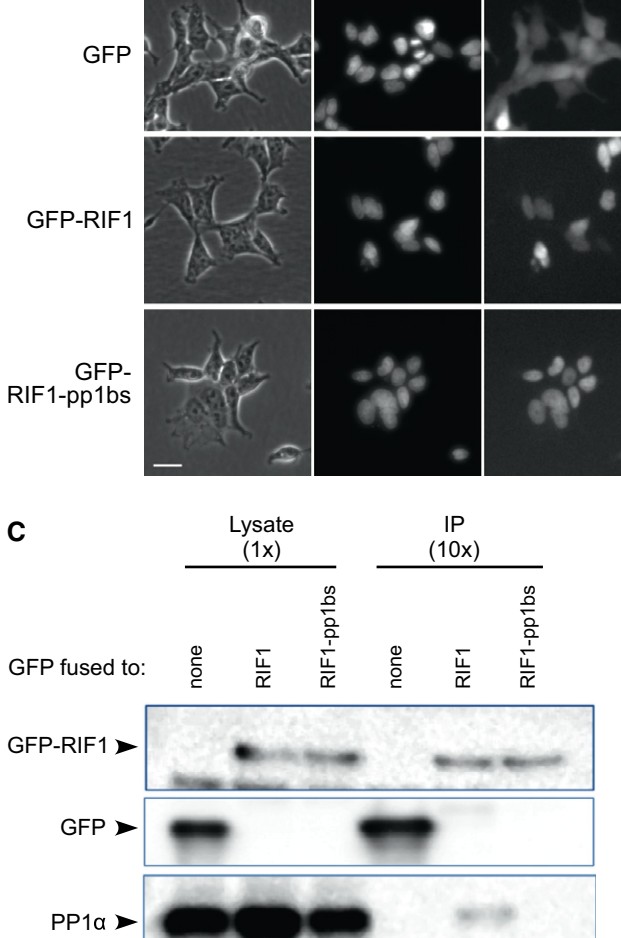

**Figure 1. RIF1 interacts with protein phosphatase 1 isoforms.**

A Construction of RIF1 cDNA mutated at its PP1 interaction motifs. Critical residues in all three potential PP1 interaction motifs are substituted with alanine, to create a RIF-pp1bs allele.

B Expression and localization of GFP-RIF1 fusion proteins in stably transfected cells. Flp-In T-REx 293 cells with GFP, GFP-RIF1, or GFP-RIF1-pp1bs were cultivated with 1 μM doxycycline (DOX) for 3 days, and expression and localization of GFP proteins were confirmed by fluorescence microscopy. Phase-contrast, DAPI-stain, and GFP images are shown. Scale bar indicates 25 μm.

C RIF1 binds PP1 protein isoforms through its PP1 interaction motifs. GFP, GFP-RIF1, and GFP-RIF1-pp1bs proteins were recovered from cell extracts using GFP-Trap beads, and co-purifying proteins were analyzed by Western blotting with anti-GFP (upper two panels) or isoform-specific PP1 antibodies (lower panels).

ectopic expression of GFP-RIF1 fully suppressed the hyperphosphorylation of both MCM4 and MCM2 (Fig 2A (ii), lane 5), demonstrating that the ectopically expressed GFP-RIF1 fusion is functional in controlling MCM phosphorylation. In sharp contrast, GFP-RIF1-pp1bs was unable to suppress hyperphosphorylation of either MCM4 or MCM2 (Fig 2A (ii), lane 7). As shown in a biological repeat of this experiment (Fig EV1), these results are generally reproducible although there is some variability in the apparent extent of effects. In particular, the apparent reduction in phosphorylated Mcm2 Ser40 in Fig 2A (ii), lane 4 is not reproducible (Fig EV1, lane 5). In general, these results imply that interaction of RIF1 with PP1 is essential to prevent hyperphosphorylation of MCM4 and MCM2.

## PP1 proteins limit MCM phosphorylation

The results above strongly suggest that RIF1 directs PP1 to dephosphorylate MCM proteins. To confirm directly that PP1 controls MCM phosphorylation, we tested the effect of PP1 depletion on MCM phosphorylation. We first depleted the individual PP1 isoforms (α, β, and γ) in HEK293 cells using isoform-specific siRNA (Fig 2B (i), lanes 1–5; Appendix Fig S1). Although siRNA treatment successfully reduced each isoform, depletion of no single isoform caused a major change in phosphorylation of MCM4 (Fig 2B (i), top panel, lanes 3–5; Appendix Fig S2A). The absence of changes in MCM4 phosphorylation could reflect functional redundancy among the PP1 isoforms. Therefore, we carried out combined depletions of the three PP1 isoforms (Fig 2B (ii), lanes 6–11; Appendix Fig S2B) and tested the effects on MCM4 phosphorylation. When PP1α and PP1γ were depleted simultaneously, phosphorylation of MCM4 increased significantly (Fig 2B (ii), lane 9), to a level close to that caused by RIF1 depletion (lane 7). Phosphorylation of MCM2 at Ser53 was also increased by PP1α and PP1γ double depletion. Triple depletion of PP1α, PP1β, and PP1γ (lane 11) caused MCM protein hyperphosphorylation similar to that caused by PP1α and PP1γ double depletion. PP1α and PP1γ may therefore be the major isoforms that regulate MCM phosphorylation levels, with PP1β making a smaller if any contribution. Overall, these results demonstrate that PP1 proteins limit the phosphorylation of MCM proteins, with PP1α and PP1γ isoforms probably playing the major role.

Taken together, the data presented imply that human RIF1 is a PP1-targeting subunit that controls MCM helicase phosphorylation.

phosphorylated MCM2 runs slightly faster in SDS–PAGE than the unphosphorylated form [37]. The results are consistent with previous observations [19] and confirm the function of RIF1 in controlling the phosphorylation status of MCM proteins. Importantly,

**A** (i)

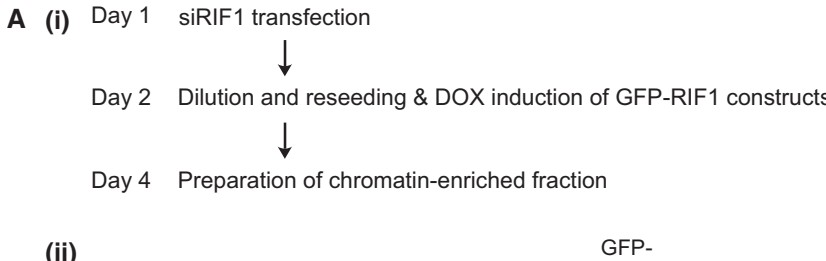

Day 1    siRIF1 transfection

Day 2    Dilution and reseeding & DOX induction of GFP-RIF1 constructs

Day 4    Preparation of chromatin-enriched fraction

(ii)

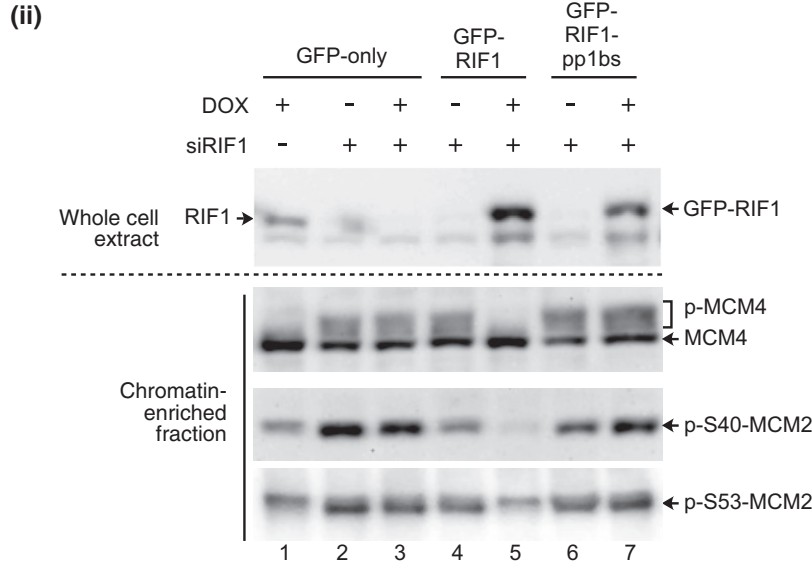

**B (i)**                                   (ii)

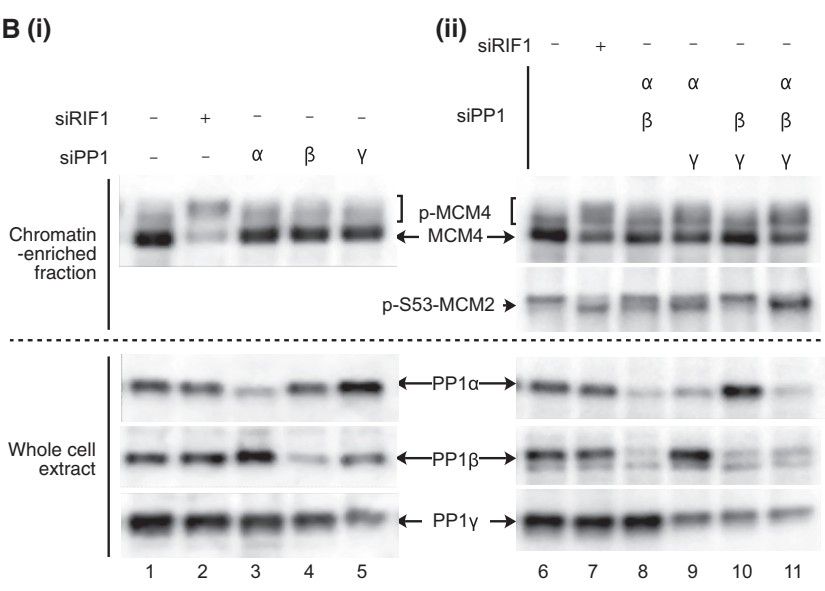

**Figure 2.  RIF1 and PP1 limit MCM phosphorylation.**

A   (i) Outline of experiment. On Day 1, Flp-In T-REx 293 cells with integrated GFP, GFP-RIF1, or GFP-RIF1-pp1bs constructs were transfected with human RIF1 siRNA or non-targeting control siRNA. On day 2, cells were diluted to ensure continuous cell proliferation throughout the experiment, and DOX was added to induce transcription of GFP, or siRNA-resistant GFP-RIF1 or GFP-RIF1-pp1bs. On day 4, chromatin-enriched samples were prepared for Western blotting. (ii) GFP-RIF1 prevents hyperphosphorylation of chromatin-associated MCM proteins, while GFP-RIF1-pp1bs cannot. Upper panel confirms removal of endogenous RIF1 and expression of GFP-RIF1 or GFP-RIF1-pp1bs. Lower three panels show Western blot analysis of chromatin-associated proteins using antibodies recognizing MCM4, phosphorylated MCM2-S40, or phosphorylated MCM2-S53. Loading was normalized by total protein as described in Materials and Methods.

B   (i) Depletion of single PP1 isoforms does not affect MCM4 phosphorylation. PP1α, PP1β, or PP1γ isoforms were depleted from HEK293 cells using isoform-specific siRNAs as indicated, and then, phosphorylation of MCM proteins was analyzed after 2 days as in (A). Depletion of PP1α, PP1β, or PP1γ was confirmed (lower three panels) by Western blotting with isoform-specific antibodies. (ii) Simultaneous depletion of PP1α and PP1γ leads to hyperphosphorylation of MCM proteins. Double and triple siRNA transfections were performed. Equal amounts of siRNAs were mixed to give a constant total siRNA concentration (50 nM), and phosphorylation status of MCM proteins were analyzed as in (A). Loading was normalized by total protein.

## RIF1 down-regulates specific phosphosites of the MCM helicase

To explore the role of RIF1 in regulating the phosphorylation status of MCM proteins, we performed comparative phosphoproteomic analysis of chromatin-associated MCM proteins from cells depleted for RIF1 or treated with control siRNA (using asynchronously growing human HEK293 cells; see Materials and Methods for details). We identified numerous ($\geq 7$) phosphorylated serine and threonine residues within each of the MCM2, MCM3, and MCM4 proteins (Table 1). Phosphorylation of MCM4 was the most affected by RIF1 depletion, with nine of the 14 sites identified showing substantially (> twofold) increased phosphorylation levels. Interestingly, the residues affected appear to be clustered (Table 1)—in particular, all five serine and threonine residues from positions 23 to 34 were identified as phosphorylated, with phosphorylation levels generally four- to sixfold increased by RIF1 depletion. A similar clustering of affected phosphorylation sites was found at MCM4 residues 131–145 (Table 1). For MCM2, RIF1 depletion led to substantially increased phosphorylation levels at three of seven identified phosphorylated residues, including serine residues at 40 and 53 consistent with the Western analysis (Fig 2). In MCM3, however, only one (Thr728) of the 10 phosphorylation sites identified was increased upon RIF1 depletion (Table 1).

We identified fewer phosphorylated residues within the other three MCM subunits, with only one or two phosphorylated residues observed for each of MCM5, MCM6, and MCM7. Among these three proteins, only one site (MCM6 Ser762) was more than twofold increased by RIF1 depletion, although several others were marginally affected (Table 1).

For all the MCM subunits, most of the phosphorylation sites affected by RIF1 depletion match either the CDK target consensus sequence [S/TP] or the DDK target consensus [S or T followed by an acidic residue, where the acidic residue may correspond to previously phosphorylated S or T] [38,39].

## RIF1 interacts with PP1 and opposes DDK activity to control replication rate

By flow cytometry analysis of cell cycle distribution, we found that ectopic expression of GFP-RIF1 leads to an accumulation of cells in S phase, consistent with a mild defect in DNA synthesis (Fig 3A). The effect is likely due to excess RIF1 protein that directs PP1 to restrict MCM phosphorylation, leading to reduced origin activation. Supporting this explanation, ectopic expression of GFP-RIF1-pp1bs caused little if any S-phase accumulation.

To further investigate the effect of RIF1 and its PP1 interaction in DNA replication, we examined DNA synthesis rate in cells ectopically expressing either GFP-RIF1 or GFP-RIF1-pp1bs. In brief, asynchronously growing cells were pulse-labeled for 15 min with 5-ethynyl-2′-deoxyuridine (EdU), and EdU incorporation analyzed by flow cytometry. Histograms (Fig 3B and C) show two cell populations: cells in S phase that are incorporating significant EdU (black plot, "EdU positive" peak), and cells at other cell cycle stages not incorporating EdU (black plot, "EdU negative" peak).

S-phase cells ectopically expressing GFP-RIF1 (and depleted for endogenous RIF1) showed reduced EdU incorporation when

**Table 1.   Changes of MCM phosphorylations caused by depleting RIF1.**

| Protein | Position | Sequence[a] | Changes in RIF1 siRNA | |
| --- | --- | --- | --- | --- |
| | | | Fold changes[b] | Log$_2$ |
| MCM2 | 26 | PLT**S**SPG | 1.30 | 0.38 |
| | 27 | LTS**S**PGR | 0.76 | −0.40 |
| | 40 | ALT**S**SPG | **2.04** | 1.03 |
| | 41 | LTS**S**PGR | 1.01 | 0.02 |
| | 53 | EDE**S**EGL | **2.55** | 1.35 |
| | 108 | LTA**S**QRE | **4.43** | 2.15 |
| | 139 | LYD**S**DEE | 0.81 | −0.30 |
| MCM3 | 163 | SDL**T**TLV | 1.29 | 0.37 |
| | 164 | DLT**T**LVA | 1.40 | 0.48 |
| | 668 | KKR**S**EDE | 0.73 | −0.46 |
| | 672 | EDE**S**ETE | 0.89 | −0.17 |
| | 674 | ESE**T**EDE | 0.88 | −0.18 |
| | 708 | YDP**Y**DFS | 1.24 | 0.31 |
| | 711 | YDF**S**DTE | 1.07 | 0.10 |
| | 713 | FSD**T**EEE | 0.92 | −0.12 |
| | 722 | QVH**T**PKT | 1.38 | 0.47 |
| | 728 | TAD**S**QET | **2.05** | 1.04 |
| MCM4 | 23 | PAQ**T**PRS | **5.63** | 2.49 |
| | 26 | TPR**S**EDA | **3.97** | 1.99 |
| | 31 | DAR**S**SPS | **5.60** | 2.48 |
| | 32 | ARS**S**PSQ | **5.60** | 2.48 |
| | 34 | SSP**S**QRR | **5.55** | 2.47 |
| | 53 | PMP**T**SPG | **2.07** | 1.05 |
| | 54 | MPT**S**PGV | 1.62 | 0.70 |
| | 61 | DLQ**S**PAA | 1.76 | 0.82 |
| | 87 | FDV**S**SPL | 0.92 | −0.13 |
| | 88 | DVS**S**PLT | 0.74 | −0.43 |
| | 120 | DLG**S**AQK | 1.42 | 0.50 |
| | 131 | DLQ**S**DGA | **2.75** | 1.46 |
| | 142 | IVA**S**EQS | **6.10** | 2.61 |
| | 145 | SEQ**S**LGQ | **7.49** | 2.91 |
| MCM5 | 498 | WDE**T**KGE | 1.69 | 0.75 |
| | 512 | TIL**S**RFD | 1.69 | 0.75 |
| MCM6 | 689 | HAD**S**PAP | 1.22 | 0.29 |
| | 762 | EID**S**EEE | **5.30** | 2.41 |
| MCM7 | 500 | PRR**S**LEQ | 1.93 | 0.95 |

[a]Relevant phosphorylated residue in bold text, shown in the context of its six surrounding residues.
[b]Changes are standardized to overall level of each protein on chromatin. Increases greater than twofold shown in bold.

compared with cells without ectopic RIF1 expression (Fig 3B, compare red plot with black and blue plots), implying a reduction in DNA synthesis rate consistent with reduced helicase activation due to excess RIF1. We wished to examine whether ectopic RIF1 expression interferes specifically with DDK-dependent steps of

**A**

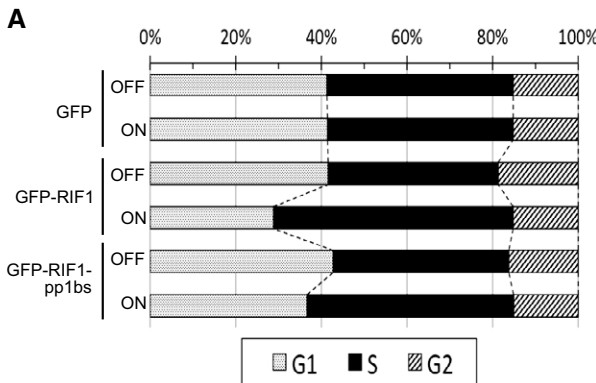

**B**

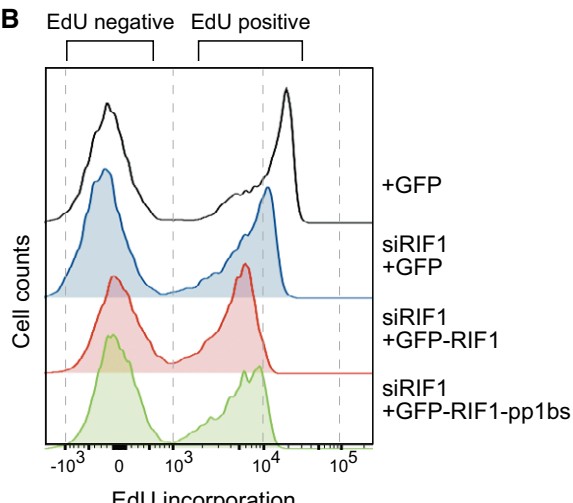

**C**

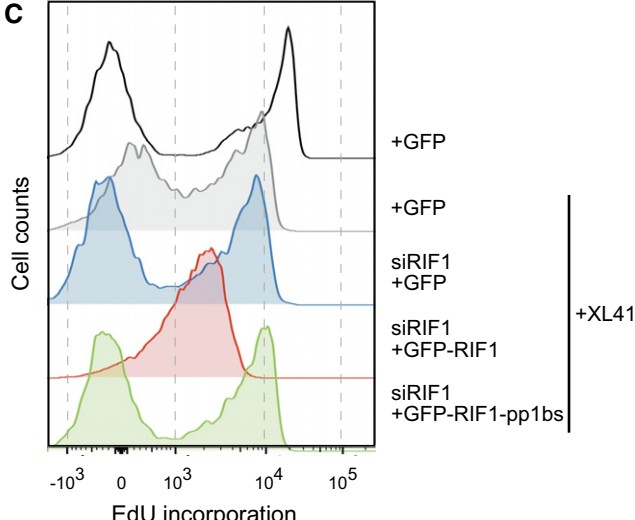

Figure 3. RIF1 affects cellular DNA synthesis.

A   Ectopic expression of GFP-RIF1 causes S-phase accumulation. Flp-In T-REx 293 cells with stable GFP, GFP-RIF1, or GFP-RIF1-pp1bs constructs were grown with or without DOX for 2 days. Plot shows cell cycle distribution of cells based on DNA content measured by flow cytometry.

B   Effect of depletion and ectopic expression of RIF1 on DNA synthesis. Flp-In T-REx 293 cells with stable GFP, GFP-RIF1, or GFP-RIF1-pp1bs constructs were depleted for endogenous RIF1 by siRNA, and then after 1 day, GFP, GFP-RIF1, or GFP-RIF1-pp1bs was induced by DOX addition. Two days later, cells were pulse-labeled with 20 μM EdU for 15 min prior to flow cytometry analysis of EdU incorporation. Overlaid histograms show EdU incorporation in each condition. Note the x-axis scale is bi-exponential.

C   Effect of depletion and ectopic expression of RIF1 on DNA synthesis in the presence of 10 μM XL413. Cells were prepared and analyzed as in (B) but with XL413 added at the same time as DOX.

Combining RIF1 ectopic expression with XL413 treatment caused a much more dramatic reduction in the EdU incorporation rate (Fig 3C, red plot), considerably more severe than the effect of either treatment alone, and indicative of substantial impairment of DNA synthesis.

These observations are confirmed by two-dimensional plots comparing EdU incorporation with overall DNA content, which give more detailed analysis of cell cycle distribution within the populations (Fig EV2). We found addition of XL413 alone caused noticeable accumulation of cells in late S phase (compare Fig EV2A (i) with EV2B (i)). This effect of XL413 is as previously described [40], and appears in the corresponding histogram as an increased population of cells incorporating a low level of EdU (Fig 3C, gray plot). The accumulation was partially rescued by depletion of endogenous RIF1 (Fig EV2B (ii)), indicating that RIF1 and DDK act in opposition to control replication. In the presence of XL413, ectopic expression of GFP-RIF1 however caused massive accumulation of cells in S phase, along with the substantial reduction in the rate of DNA synthesis as measured by EdU incorporation (Fig EV2B (iii)). This aberrant profile suggests that ectopic RIF1 expression combined with DDK inhibition impairs replication rate to such an extent that the cells spend most of the cell division cycle duplicating their DNA.

The replication rate was however not reduced by ectopic expression of GFP-RIF1-pp1bs along with XL413 addition (Fig 3C, green plot, and Fig EV1B (iv)), which produced profiles resembling those caused by control GFP expression. RIF1 therefore requires PP1 interaction to limit DNA synthesis rate and oppose DDK activity.

We found moreover that an inhibitory effect of XL413 on cell proliferation was enhanced in cells overexpressing GFP-RIF1, but not GFP-RIF1-pp1bs (Appendix Fig S3). These results are consistent with those described in yeast, where the effect of mutating DDK components is relieved by loss of *RIF1* [11–13,18].

To summarize, the removal of RIF1 can relieve inhibitory effects of XL413 on DNA synthesis, showing that RIF1 and DDK act in opposition to control replication. Conversely, ectopic expression of GFP-RIF1 limits DNA synthesis rate in a way that is strongly synergistic with DDK inhibition. This effect of RIF1 depends on its interaction with PP1, so that overall, the effects of RIF1 on MCM phosphorylation, and the associated requirement for DDK activity, are strikingly similar to those described in yeast.

replication, as is the case in budding yeast [11–13]. We therefore tested whether simultaneously limiting DDK activity exacerbates effects of ectopic RIF1 expression, by combining ectopic RIF1 expression with addition of the DDK inhibitor XL413 [40]. XL413 treatment alone caused some reduction in EdU incorporation, reflecting repression of cellular DNA synthesis rate due to impaired DDK activity (Fig 3C, compare black and gray histogram profiles).

## RIF1-PP1 is required for full origin licensing

Overexpression of GFP-RIF1 negatively impacted DNA synthesis rate, especially in the context of DDK inhibition (Fig 3C), confirming that human RIF1-PP1 controls DNA replication by counteracting DDK-mediated MCM phosphorylation (Fig 2). If RIF1 acts simply as a repressor of replication, RIF1 removal would be expected to accelerate replication rate and potentially S-phase progression. However, we noticed that depletion of RIF1 (in the absence of XL413) in fact reduced the cellular DNA synthesis rate (Fig 3B, compare black and blue profiles). This cannot be explained by RIF1-PP1 acting only to limit MCM phosphorylation, but rather suggests that RIF1 additionally has a novel, positive function in stimulating DNA replication.

In our proteomic analysis, we noticed that the overall loading of MCM2–7 proteins on chromatin was reduced in cells depleted for RIF1 (Fig 4A). Importantly, all six MCM subunits were uniformly reduced (by ~50% relative to control), indicating that chromatin association of the entire MCM complex is compromised by RIF1 removal. This reduced overall MCM loading on chromatin stands in contrast to the increased phosphorylation levels of certain sites on the chromatin-associated MCM subunits (Fig 4B). To investigate further, we analyzed chromatin-associated MCM3 protein in relation to cell cycle stage, using a quantitative flow cytometry method [41]. This approach is suitable for analyzing unperturbed cell populations to identify cell cycle-related changes that would be difficult to detect by Western blotting. In brief, cells were treated with a low concentration of non-ionic detergent to remove soluble proteins before fixation, and then, the detergent-resistant proteins were detected by indirect immunofluorescence. This approach is analogous to extraction methods used in preparation of chromatin-enriched protein fractions.

Following soluble protein extraction, cells were analyzed by flow cytometry for abundance of MCM3 protein and simultaneously for

DNA content (Fig 4C). As described by Haland *et al* [41], the observed pattern of detergent-resistant MCM3 in control cells is consistent with the established behavior of MCM proteins during

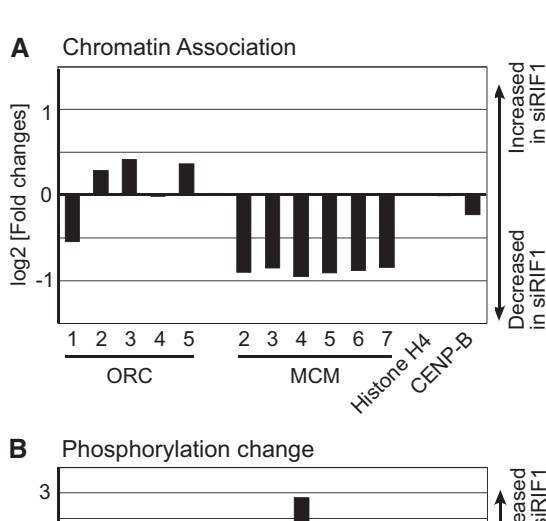

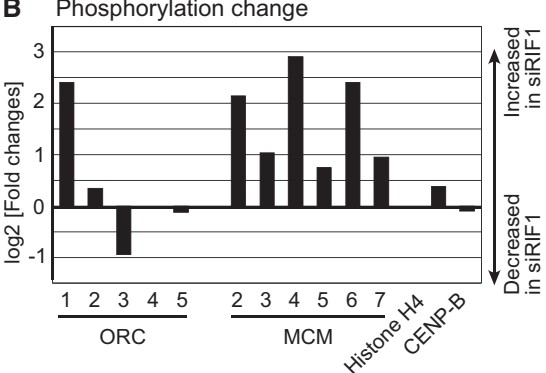

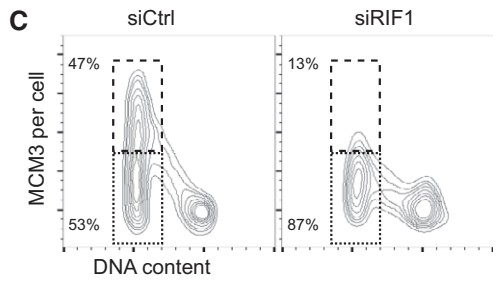

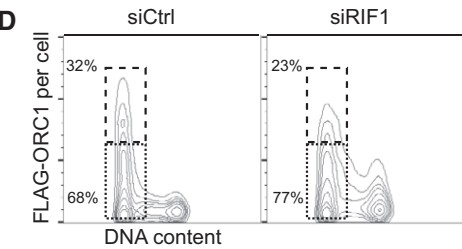

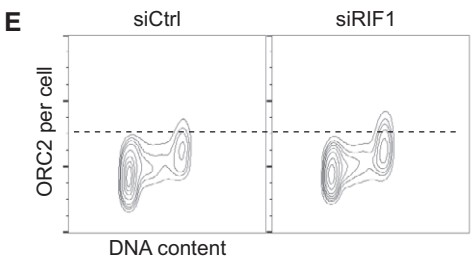

---

**Figure 4. RIF1 is required for full replication licensing.**

A  Reduced chromatin association of ORC1 and MCM proteins in RIF1-depleted cells. Amounts of chromatin-associated ORC and MCM proteins in HEK293 cells treated with siCtrl or siRIF1 were analyzed by quantitative proteomics as in Table 1. ORC6 was not detected perhaps due to its weak association with other ORC subunits [75]. Histone H4 and CENP-B are controls showing similar chromatin association in the presence or absence of RIF1.

B  Increased phosphorylation of ORC1 and MCM proteins in RIF1-depleted cells. Relative changes in phosphorylation of chromatin-associated ORC and MCM proteins were analyzed as in Table 1. Plot shows the log$_2$ value for the residue most affected by depleting RIF1, for each protein. Values are normalized by the change in chromatin association of each protein.

C  MCM loading onto chromatin is impaired in RIF1-depleted cells. Abundance of chromatin-associated MCM3 in control and RIF1-depleted HEK293 cells was analyzed by flow cytometry. Two-dimensional contour plots are shown (x-axis: DNA content, y-axis: MCM3 protein abundance). The y-axis is on a linear scale. Dashed and dotted boxes indicate G1 cells with "high-MCM3" and "low-MCM3", respectively.

D  Reduced chromatin association of ORC1 in RIF1-depleted cells. Amounts of chromatin-associated FLAG-ORC1 in control and RIF1-depleted FLAG-ORC1 HEK293 cells were analyzed by flow cytometry as in (C). Dashed and dotted boxes indicate G1 cells with "high-FLAG-ORC1" and "low-FLAG-ORC1", respectively.

E  ORC2 was similarly analyzed, in the same set of cells as in (D).

the cell cycle: MCM3 is loaded onto the chromatin in G1 phase and gradually dissociates during S phase concurrent with progress of DNA replication, arriving at a minimum for cells in G2/M phase (Fig 4C, left).

In the cells depleted for RIF1, MCM3 association with chromatin during G1 phase was reduced (Fig 4C, right), with the maximum MCM3 level in RIF1-depleted cells ("top end" of the G1 population) much lower than for control cells. In siCtrl cells, the ratio of "high-MCM3" to "low-MCM3" cells (dashed and dotted boxes, respectively) was almost 1:1 with 47% of cells in the "high-MCM3" category. In contrast, only 13% of RIF1-depleted cells displayed a "high-MCM3" load on chromatin (Fig 4C). RIF1 depletion itself does not significantly affect cell cycle distribution (data not shown), and so, this reduced MCM loading is not due to premature initiation of S phase. Together with the mass spectrometry data, these data demonstrate that MCM loading on chromatin is significantly impaired in RIF1-depleted cells, indicating a positive role of RIF1 in origin licensing during G1 phase. The reduction of chromatin-associated MCM4 protein caused by RIF1 depletion was confirmed to be reproducible in Western blotting experiments (Fig EV3A). Moreover, ectopic expression of GFP-RIF1, but not GFP-RIF1-pp1bs, rescued this reduced MCM4 chromatin association, implying that interaction of RIF1 with PP1 is important to ensure full licensing (Fig EV3A).

## RIF1-PP1 protects ORC1 protein from proteasome-dependent destruction in G1 phase

Seeking the mechanism through which RIF1-PP1 might regulate origin licensing, we noticed a modest reduction of ORC1 protein in the chromatin-enriched fraction of RIF1-depleted cells (Fig 4A). Other ORC subunits are in contrast either unchanged, or slightly increased (Fig 4A). Human ORC1 protein is known to behave differently from other ORC proteins: while other ORC subunits remain associated with chromatin throughout the cell cycle, ORC1 is stable and chromatin-associated only during G1 phase [5,6,42]. Outside of G1 phase, cyclin A–CDK2 phosphorylates ORC1, and this phosphorylation triggers ORC1 degradation by the proteasome [5,6,42], though the precise phosphorylation sites responsible are not yet determined. Importantly, ORC1 protein is therefore proposed to be a limiting factor for origin licensing [5].

Based on our observation of reduced ORC1 chromatin association upon RIF1 depletion, we hypothesized that RIF1-PP1 protects ORC1 from destruction in G1 phase by ensuring that it is kept unphosphorylated. In this model, loss of RIF1 would result in untimely phosphorylation and degradation of ORC1, and a consequent defect in MCM loading, as observed (Fig 4C). To test this hypothesis, we assessed the effect of depleting RIF1 on the abundance of chromatin-associated ORC1 (Fig 4D) using a HEK293-derived cell line that ectopically expresses FLAG-tagged ORC1 [6]. Importantly, this FLAG-ORC1 protein is subject to cell cycle-controlled degradation and shows abundance and chromatin association fluctuation during the cell cycle like endogenous ORC1 [6,42]. Both endogenous ORC1 and FLAG-ORC1 proteins are almost exclusively associated with chromatin in G1 phase [5,6,42], so that the abundance of chromatin-associated FLAG-ORC1 mirrors cellular abundance of ORC1 protein.

Using a "protein extraction-flow cytometry" approach similar to that described for MCM3, we found that FLAG-ORC1 protein is hardly present in G2 cells and becomes chromatin-associated only in G1 phase (Fig 4D, left), with 32% of Control G1 phase cells showing high levels of ORC1 chromatin association. The amount of FLAG-ORC1 that becomes chromatin-associated was reduced in RIF1-depleted cells, with only 23% of G1 phase cells showing high-level association (Fig 4D, right), consistent with the suggestion that RIF1 is needed to stabilize ORC1. The reduction was even greater upon depletion of RIF1 from HeLa cells expressing GFP-fused ORC1 and mCherry-PCNA (Fig EV3B–D), with the proportion of G1 phase cells showing highest ORC1 chromatin association dropping from 27 to 13% (differing transfection efficiency may account for the higher impact of siRIF1 in this cell line compared to that in Fig 4D). We confirmed moreover that MCM3 chromatin association is also reduced when RIF1 is depleted in the FLAG-ORC1 cells (Fig EV3E). In contrast to ORC1, we found that ORC2 protein remains chromatin-associated stably throughout the cell cycle (Fig 4E, left panel), consistent with previous studies [5,42]. ORC2 chromatin association was unaffected by RIF1 depletion (Fig 4E, right panel), in agreement with our proteomics data (Fig 4A). Overall, these results demonstrate that RIF1 is required for efficient stabilization of the ORC1 protein and its chromatin association during G1 phase.

If RIF1 acts with PP1 to protect ORC1 from degradation, then PP1 inhibition should increase ORC1 phosphorylation leading to its degradation. To test this prediction, we analyzed ORC1 abundance when PP1 activity is inhibited. Cells expressing FLAG-ORC1 were treated with the PP1 inhibitor tautomycetin [43] for 4 h and the ORC1 protein abundance was analyzed (Fig 5A). FLAG-ORC1 abundance and chromatin association were substantially reduced in cells treated with tautomycetin (Fig 5A, right), and chromatin association of MCM3 was also reduced in the same cells (Fig 5B, right). Note that 4-hr treatment with 5 μM tautomycetin did not noticeably change cell cycle distribution (Fig EV3F); therefore, the changes in ORC1 and MCM3 levels are not due either to impaired exit from mitosis, or to other cell cycle defects. The effects of tautomycetin indicate that, along with RIF1, PP1 activity is important to protect ORC1 protein from degradation, and is essential for full replication licensing in human cells.

To confirm that RIF1 protects ORC1 from degradation by the proteasome, we analyzed the abundance of FLAG-ORC1 when proteasomal degradation is inhibited. If the reduced ORC1 in RIF1-depleted cells is due to accelerated proteasome-mediated degradation, proteasome inhibition should rescue the reduction in ORC1 protein level. To test this, we treated control and RIF1-depleted cells with proteasome inhibitor MG-132 for 4 h prior to analysis by flow cytometry. Incubation of control cells with MG-132 slightly increased FLAG-ORC1 abundance in G1 cells, as well as at other cell cycle stages (Fig 5C), indicating that some ORC1 protein is proteasomally degraded even in the presence of RIF1. In RIF1-depleted cells, proteasome inhibition partly stabilized FLAG-ORC1 protein in G1 as well as other cell cycle phases (Fig 5C), such that the maximum FLAG-ORC1 level in siRIF1 cells treated with MG-132 is comparable to that in control cells. In the GFP-ORC1-expressing HeLa cell line, G1 phase destabilization of GFP-ORC1 due to RIF1 depletion was fully rescued by MG-132 treatment (Fig EV3G). Also with this cell line, proteasome inhibition led to increased ORC1 at other cell cycle

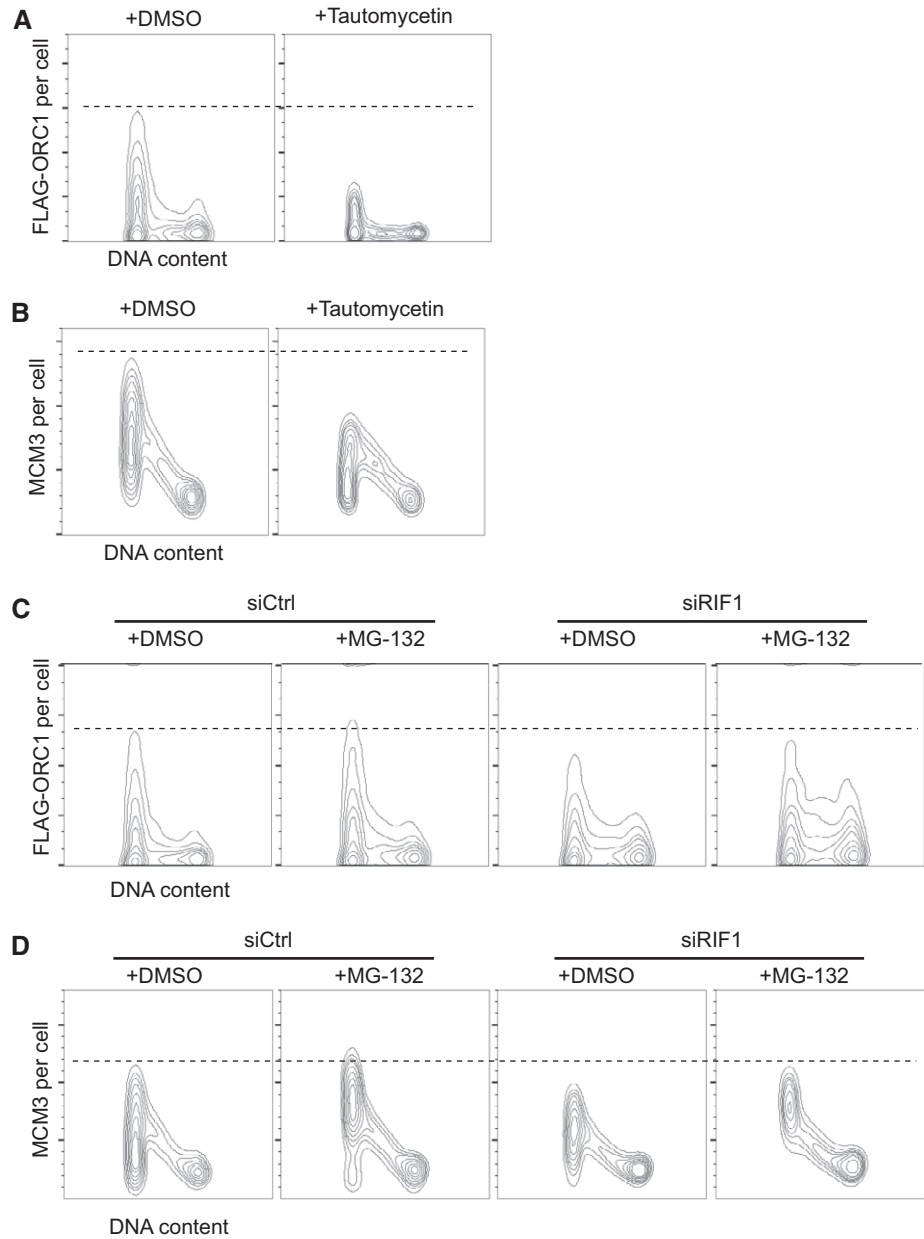

**Figure 5.   RIF1 and PP1 protect ORC1 from cell cycle-specific degradation during G1 phase.**

A   Inhibition of PP1 destabilizes ORC1. FLAG-ORC1 HEK293 cells were treated for 4 h with 5 μM tautomycetin prior to flow cytometry, and chromatin-associated FLAG-ORC1 analyzed as in Fig 4D.

B   Inhibition of PP1 causes reduced MCM loading. Chromatin-associated MCM3 protein was analyzed in the same set of the cells as in (A).

C   Reduced ORC1 chromatin association in RIF1-depleted cells is partly rescued by inhibiting the proteasome. Control and ORC1-depleted FLAG-ORC1 HEK293 cells were treated for 4 h with 20 μM MG-132 (or DMSO) prior to flow cytometry analysis.

D   Proteasome inhibition rescues MCM3 chromatin loading. Chromatin-associated MCM3 was analyzed in the same set of cells as in (C).

phases as expected. Chromatin association of MCM3 protein was assessed in the FLAG-ORC1 cell line in the same experiment (Fig 5D). Incubation of cells with MG-132 led to a clear rescue of the reduced MCM3 chromatin association caused by siRIF1, to a level comparable to control cells (Fig 5D). Overall, these results are consistent with our proposal that RIF1-PP1 acts to stabilize ORC1 protein and protect it from proteasomal degradation, hence promoting MCM loading during G1 phase.

**Reduced origin licensing due to Rif1 depletion leads to increased origin spacing**

As described above, we found a modest but reproducible reduction of origin licensing in RIF1-depleted and in PP1-inhibited cells (Figs 4 and 5). In normal cells, considerably more sites are loaded with MCM proteins than are required to complete chromosomal DNA replication, so that a substantial number of

licensed origins remain "dormant" in an unchallenged S phase [44,45]. We investigated the effect of Rif1 on frequency of origin initiation in unperturbed S phase, and also in cells treated with hydroxyurea (HU) which interrupts replication fork progression and stimulates dormant origin activation [45]. Interorigin distance (IOD) was analyzed using the DNA fiber technique (Fig 6A) in control and RIF1-depleted cells. Without HU (Fig 6B), the IOD in siCtrl cells ranges from around 30 to 130 kb, with a median of 63.0 kb (Fig 6B, open bars). RIF1-depleted cells show a broader range of IOD values with a statistically significant increase in the median value to 77.0 kb ($P = 0.001$), and an increased incidence of origin spacings above 120 kb (Fig 6B, black bars). The spacing of active origins is therefore increased by Rif1 depletion, consistent with a reduction in licensing. Upon treating siCtrl cells with 200 μM HU, activation of dormant origins occurs, as evidenced by a noticeably increased proportion of IOD values shorter than 40 kb (Fig 6C, compare open bars in pink shaded region with open bars in Fig 6B), and a reduction in the median IOD to 45.9 kb. RIF1-depleted cells treated with HU showed noticeably fewer IOD values < 40 kb (Fig 6C, black bars within pink shaded region), and a longer IOD median (55.2 kb). Removal of RIF1 therefore also compromises the availability of dormant origins for activation upon HU treatment. The increase in median IOD caused by RIF1 depletion both in the presence and absence of HU confirms that RIF1 removal causes an overall decrease in the total number of origins that initiate replication. In agreement, RIF1 depletion also led to a decrease in the overall DNA synthesis rate in the presence and absence of HU (Figs 3B and EV4A). The speed of progress of replication forks was however not significantly changed by RIF1 depletion (Fig EV4B). These observations are generally consistent with a requirement for human RIF1 to mediate full origin licensing (Figs 4–6), in addition to its role in limiting MCM phosphorylation levels (Fig 2 and Table 1).

## Discussion

While RIF1 was known to affect DNA replication throughout eukaryotes, it has been unclear whether its molecular mechanism of action is conserved, partly because of the complex replication phenotypes associated with loss of mammalian RIF1. Our studies demonstrate that human RIF1 can interact with all three PP1 isoforms and is important to prevent excessive phosphorylation of MCM proteins (Figs 1 and 2, and Table 1). This function parallels the molecular action of yeast RIF1 in limiting origin initiation, and demonstrates that PP1 targeting is a central molecular function of RIF1 conserved throughout evolution. We identified phosphorylation sites on human MCM proteins affected by RIF1, and quantitatively assessed the impact of depleting RIF1 (Table 1). Although PP1 itself has poor substrate specificity, phosphorylated residues affected by RIF1 appear confined to particular regions, notably the regulatory N-terminal domains of MCM2 and MCM4. This suggests that RIF1 may direct PP1 to dephosphorylate specific regions of the MCM complex. Interestingly, although MCM2 and MCM4 are located on the opposite sides of a single MCM hexamer, their N-termini are juxtaposed in the MCM double hexamer formed during origin licensing [46]. Further structural studies will be needed to

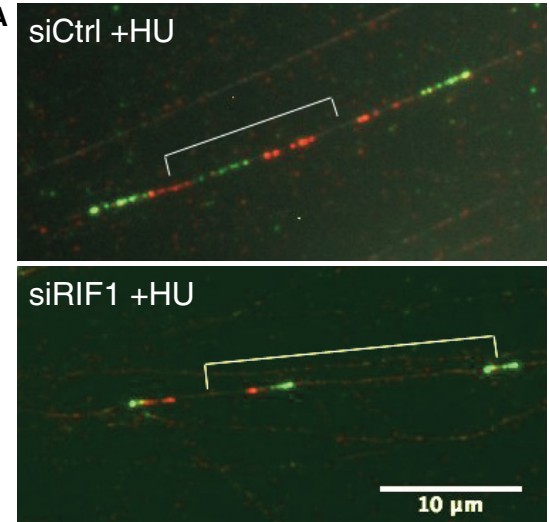

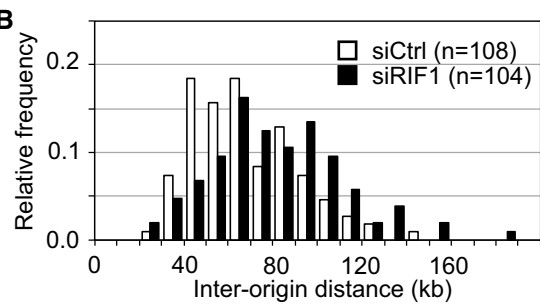

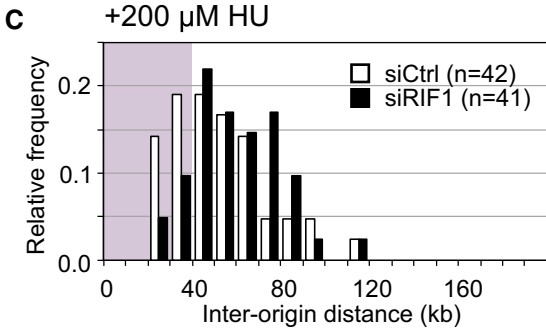

**Figure 6.  Effect of RIF1 on origin activation in unperturbed S phase.**

A  Specimen images of DNA fibers sequentially labeled with CldU and IdU. siControl- and siRIF1-treated Flp-In T-REx 293 cells (with integrated GFP) were incubated for 4 h in the presence of 200 μM HU, with pulse-labeling using CldU (visualized red) then IdU (visualized green) in the final 40 min.

B  Distribution of origin distance in siCtrl (open bars) and siRIF1 (black bars) cell lines, treated as described in (A) but with no HU treatment. For statistical analysis of IOD data, Mann–Whitney–Wilcoxon test was performed using R (version 3).

C  Distribution of origin distance in siCtrl (open bars) and siRIF1 (black bars) cell lines, treated as described in (A) including HU treatment. The area corresponding to IOD values shorter than 40 kb is marked by pink background.

understand how particular domains of the MCM complex are specifically affected by RIF1-PP1.

Analysis of EdU incorporation (Fig 3) suggested that RIF1 might have other effects on DNA replication control beyond simply acting

as a suppressor of initiation. In particular, depletion of endogenous RIF1 leads to reduced DNA synthesis, suggesting that RIF1 might positively regulate replication. Further investigation revealed a previously unappreciated role for RIF1 in ensuring optimal replication licensing. Along with PP1, RIF1 contributes to the G1-specific stabilization of ORC1, ensuring that cells are fully competent for MCM loading and pre-RC assembly at the correct cell cycle phase. The fact that RIF1 depletion causes only a mild defect in ORC1 loading but has a somewhat larger impact on MCM loading (Fig 4A, C and D) is as expected, since under normal circumstances, multiple MCM complexes are believed to be loaded at each single ORC site. Overall, effects of RIF1 that previously seemed paradoxical can be explained by the discovery that RIF1-PP1 has two functions in replication control as illustrated in Fig 7: first, the promotion of origin licensing by protecting ORC1 from degradation, and second, suppression of origin firing by counteracting DDK.

We found that either loss of RIF1, or inhibition of PP1, leads to an almost twofold reduction in replication licensing levels as measured by MCM association with chromatin in proteomic and flow cytometry experiments (Figs 4 and 5). The magnitude of this reduction is comparable to the drop in DNA synthesis rate in RIF1-depleted S-phase cells, which incorporate EdU at only about 58% of the normal rate (Figs 3B and EV4A). RIF1 depletion does also cause a fairly mild increase in interorigin distance, our fiber analyses showing an increase of 10–20 kb in typical origin spacing. While consistent with the reduced replication licensing, this decrease in the density of initiating origins is not sufficient to fully account for the drop in DNA synthesis rate caused by RIF1 depletion.

How can loss of RIF1 substantially compromise replication licensing and DNA synthesis rate, while only mildly increasing

IOD? One potential explanation may be that some of the effects of RIF1 on replication licensing operate at a broader level than is detectable by fiber analysis [17,19,22]. Fiber analysis measurements of IOD are effective for identifying changes occurring at a local level, but changes in replication dynamics that involve entire chromosome domains cannot be visualized through fiber measurements. For example, if RIF1 depletion only mildly affects licensing of most regions, but in some areas causes failure to license entire clusters of replication origins, then the effect will be to compromise the overall replication rate more than is reflected by local IOD measurements (as group of origins that failed to license are invisible in fiber analysis). RIF1 loss is known in some contexts to cause large-scale effects on groups of origins: for example, RIF1 removal affects the replication timing program of human and mouse cells such that entire domains—encompassing hundreds of kilobases and containing multiple replication origins—show substantially altered replication timing [17,19].

The exact molecular mechanism through which RIF1 stabilizes ORC1 will require further investigation, especially since the phosphorylation site(s) directing degradation of ORC1 is yet to be identified. While we cannot completely rule out the possibility of an indirect effect, it is interesting to note that the ORC1 Ser273 residue, which showed significantly increased phosphorylation on RIF1 depletion in our proteomic analysis, lies close to a potential destruction box at residues 236–244, and to a PP1 interaction motif at residues 264–268. PP1 interaction motifs are characteristic of PP1 substrates as well as PP1-targeting subunits, so this motif could potentially mediate PP1 docking to dephosphorylate Ser273 and stabilize the ORC1 protein.

Two other proteins required for origin licensing in human cells, Cdc6 and Cdt1, are also regulated through phosphorylation by CDK2. Phosphorylated Cdc6 is exported from the nucleus, while phosphorylation of Cdt1 targets it for degradation [47,48]. We tested whether these proteins might also be regulated by RIF1. Cdc6 localization was however not altered by RIF1 depletion (Fig EV5A), while any effect of RIF1 on Cdt1 stability appears very slight (Fig EV5B).

## Replication timing

Removal of either human or mouse RIF1 causes large-scale switches in the replication time of extended chromosome domains [17,19]. A recent paper used chromatin immunoprecipitation (ChIP) to demonstrate that in mouse embryonic stem (ES) cells, RIF1 is bound mainly to a subset of late-replicating regions, and that the same regions tend to display earlier replication timing in cells deleted for RIF1. How can this apparently domain-specific role for RIF1 in replication timing be reconciled with the effects on licensing and origin activation described here? Resolving this issue clearly requires further investigation, but one possibility is that RIF1 interacts with different classes of chromosomal domains at various cell cycle stages to mediate its distinct functions. During mitosis, RIF1 is dissociated from chromatin but re-associates as cells re-enter G1 phase. The early G1 phase chromatin association has been investigated only microscopically but appears to be general [17,19,25], consistent with the possibility that RIF1 may associate genomewide at the right time to support the establishment of licensed origins. Once cells enter S phase, however, RIF1 has been microscopically

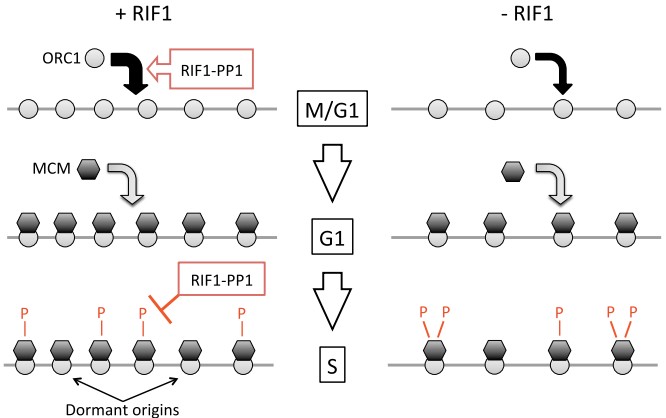

**Figure 7. Model of DNA replication control by RIF1-PP1.**
In normal cells (left panel), RIF1 acts with PP1 to stabilize ORC1, promoting its loading at potential DNA replication origin sites at M/G1 phase (top). ORC1 then stimulates MCM loading, to license origins in G1 phase (middle). When cells enter S phase, RIF1 counteracts DDK-mediated phosphorylation of the MCM complex to limit the number of origins activated (bottom). In the absence of RIF1 (right panel), ORC1 is short-lived and its chromatin association reduced (top), so that fewer origins are licensed in G1 (middle). During S phase, however, increased MCM phosphorylation means that a larger fraction of licensed origins are activated (bottom). For simplicity, the ORC2–5 subunits (which are chromatin-associated throughout the cell cycle) are omitted.

described as showing a more limited binding pattern, associating mainly with unreplicated DNA in heterochromatic, late-replicating regions [17,19,25], consistent with the pattern identified by ChIP in ES cells [22]. In this configuration, RIF1 would be positioned such that it could delay replication by preventing MCM helicase activation until the right time in S phase. If RIF1 does change its chromatin association as the cell cycle progresses, then it is possible that the ChIP localization of RIF1 reported by Foti *et al* [22] in ES cells mainly represents the S-phase pattern, since ES cells appear to spend most of interphase replicating their DNA [49,50].

The model outlined above proposes that in early G1 Rif1 associates genomewide and supports replication licensing, and then later once cells enter S phase, it dissociates specifically from early domains and is left only at some late domains where it delays replication. While this suggestion appears to fit with most previous observations, other models are of course possible. In particular, there is compelling evidence that mouse RIF1 mediates interdomain organization during G1 phase, potentially impacting on the replication timing program [22].

A further possibility is that RIF1 affects the replication timing program by ensuring "effectiveness of licensing" of replication origins. Multiple MCM complexes can be loaded per ORC complex [51], and it is suggested that the number of MCM complexes at an origin determines the timing and the efficiency of its activation [52,53]. Reduced ORC1 half-life in the absence of RIF1 may shorten the window of opportunity for MCM loading at each origin, potentially resulting in a smaller number of MCM complexes loaded, which would lead to an apparent reduction in licensing as observed in our results. This would in turn disturb the replication program, if MCM complex number at each origin is indeed the major determinant of replication timing.

It should be noted that the possibilities outlined above are not incompatible with each other, and that these mechanisms may function either independently, or in concordance so that RIF1 acts through several mechanisms each contributing to sharpen the replication timing program.

### Dormant origins and damage sensitivity

The local activation of dormant origins upon replication stress—for example, due to inhibition of replication fork progression—is thought to protect cells from replication inhibitors, presumably by providing a "backup" mechanism to replicate the DNA when replication forks irreparably stall or collapse [54–56]. We found that Rif1 depletion leads not only to an increase in origin spacing in normal S phase, but also to a reduction in the availability of dormant origins following replication stress. One consequence of limiting the number of available dormant origins is increased sensitivity to replication-inhibiting drugs, such as hydroxyurea or aphidicolin. Interestingly, cells lacking RIF1 are very sensitive to both hydroxyurea and aphidicolin [25]. We suspect that the hypersensitivity of cells lacking RIF1 to replication inhibitors may reflect the fact they have fewer dormant origins available to respond effectively. Overall, the dual function we have identified for RIF1 in replication control—stimulating replication licensing but repressing origin activation—may explain why simple removal of RIF1 allows unimpeded replication to progress without obvious problems, but leaves the cells

more sensitive to disruption of affected pathways. Thus, the vulnerability and fragility of replication control caused by deregulation of RIF1 are exposed either by interrogating dormant origin availability (using aphidicolin or hydroxyurea), or by compromising MCM phosphorylation (by DDK inhibition with XL413).

### Is RIF1 function in origin licensing conserved?

It is unclear whether the function of RIF1 in origin licensing is conserved. Chromatin association of hamster ORC1 is controlled at least in part through regulated cell cycle localization [57], and it remains to be investigated whether either the localization or stability of rodent licensing components is regulated by RIF1. In yeast, ORC1 is associated with DNA replication origins throughout the cell cycle, and there is no clear evidence of a role for yeast Rif1 in supporting licensing.

### Other functions of RIF1 and ORC1

ORC1 has been implicated in a number of cellular functions in addition to replication licensing, most notably centrosome replication and as a consequence cilia formation [58–60]. Further investigation will be needed to ascertain whether reduced ORC1 protein in RIF1-deficient cells causes defects in centrosome control and cilia formation. Origin licensing proteins, including ORC1, are linked to Meier-Gorlin syndrome [59,61]. Our discovery that RIF1 supports licensing through ORC1 therefore raises the intriguing possibility that RIF1 mutations could potentially also contribute to Meier-Gorlin syndrome.

Mammalian RIF1 affects various pathways of DNA damage recognition and repair [13]. RIF1 is recruited to DSB sites through interaction with 53BP1, where it directs cell cycle-dependent DSB repair pathway choice [34,62–64]. RIF1 also acts in resolution of ultrafine DNA bridges that may result from replication fork failure [65]. Our demonstration here that human RIF1 is a PP1-targeting subunit raises the possibility that other functions of RIF1—in particular repair pathway choice—are mediated by controlling phosphorylation of components of the network that directs DNA repair.

## Materials and Methods

### PCR primers

The following PCR primers were used to mutagenize the PP1 interaction motifs of human RIF1 (mutated nucleotides in lower-case letters):
SH568: 5′-GCTTTCAAAATTGAATGATACCATTAAGAATTCAG-3′
SH569: 5′-GCAATggcAGCggcCTTTTTAATCATGGGTGCTCCAC-3′
SH570: 5′-GgccGCTgccATTGCTTGGAAGAGTTTAATAGATAATTTTGC-3′
SH571: 5′-CAGGTTCATTCGGGGAGTTCCC-3′
SH572: 5′-CTATGGAATTGAATGTAGGAAATGAAGCTAGC-3′
SH573: 5′-CAGGTGATGAGATTTCATCTTCTTGGGATCTTTTTAGTCCTCTCTTggcggcGCTCGTAGACGGAGAAGCCAAA-3′
SH574: 5′-GAAGATGAAATCTCATCACCTGTTAATAAGGTTCGCCGTGcCTCCgccGCAGATCCAATATACCAAGCAGGA-3′
SH575: 5′-CGAGTCTTCACCTGCTGCTCATGATATATTCTG-3′

## Plasmids

The GFP-RIF1 constructs used in this study are based on pcDNA5/FRT/TO-GFP-RIF1 [34], which carries human RIF1 cDNA fused to GFP at its N-terminus. The cDNA corresponds to the shorter of two reported splicing variants [66], and encodes a 2,446-amino acid protein (corresponds to Q5UIP0-2 in the Uniprot protein database). To mutagenize the N-terminal PP1 interaction motif of RIF1, the *Eco*RI-*Xma*I fragment of plasmid pcDNA5/FRT/TO-GFP-RIF1 was replaced using the In-Fusion HD cloning system (Clontech) by two PCR fragments amplified with primers SH568 & SH569, and SH570 & SH571, respectively. To mutagenize the C-terminal PP1 interaction motifs of RIF1, the *Nhe*I-*Aar*I fragment of the plasmid was similarly replaced by two PCR fragments amplified with primers SH572 & SH573, and SH574 & SH575, respectively. Introduction of the designed mutations and the absence of unexpected mutations was confirmed by DNA sequencing.

## Cell lines and culture conditions

Cells were cultivated in Dulbecco's modified Eagle's minimal medium supplemented with 10% fetal calf serum at 5% $CO_2$ and ambient $O_2$ at 37°C. Tetracycline-free fetal calf serum (Labtech, FB-1001T/500) was used for cultivation of cell lines with tetracycline-inducible constructs. Appropriate antibiotics were added for selection of the constructs.

The Flp-In T-REx 293 cell line (Invitrogen) was used to create stable cell lines for the ectopic expression of GFP, GFP-RIF1, and GFP-RIF1-pp1bs. To construct cell lines, pOG44 [67] and the pcDNA5/FRT/TO-based plasmids carrying GFP, GFP-RIF1, or GFP-RIF1-pp1bs genes were mixed in 9:1 molar ratio and used for transfection of Flp-In T-REx 293 cells with Lipofectamine 3000 (Invitrogen). Transfections and hygromycin B selection of stably transfected cells were performed as described by the manufacturer. Clones were tested for doxycycline-dependent induction of GFP fusion proteins by Western blots and microscopy.

To assess the effect of depleting and ectopically expressing RIF1, cells were transfected with either control siRNA or siRNA against human RIF1. The following day, cells were split and cultivated for a further 2 days with 1 μM DOX to induce GFP, GFP-RIF1, and GFP-RIF1-pp1bs proteins. Where noted, XL413 was added at 10 μM together with DOX.

The stable HEK293-derived cell line expressing FLAG-ORC1 was previously described [6]. The stable HeLa-derived cell line expressing FLAG-GFP-ORC1 mCherry-PCNA will be described elsewhere more in detail.

U2OS and HEK293 cell lines were previously described [68,69] and were obtained from Public Health England.

## List of siRNA used in this study

1  Human RIF1 siRNA (Dharmacon, D-027983-02).
2  Human PP1α siRNA (Santa Cruz Biotechnology, sc-36299).
3  Human PP1β siRNA (Santa Cruz Biotechnology, sc-36295).
4  Human PP1γ siRNA (Santa Cruz Biotechnology, sc-36297).
5  Control siRNA against luciferase (Dharmacon, D-001100-01).

## Chromatin fractionation and immunoprecipitation

Chromatin-enriched protein fractions were prepared essentially as described [70]. Protein concentrations in chromatin-enriched fractions were determined using the Bio-Rad RC DC Protein assay kit. For Western blots, equal amount of total proteins were loaded on each lane, and for quantification, the results were generally further normalized by total protein using Bio-Rad stain-free gels. When stain-free normalization was unavailable, equal loading on the blot was confirmed by Ponceau S staining.

Immunoprecipitation of GFP, GFP-RIF1, and GFP-RIF1-pp1bs proteins was carried out using GFP-Trap beads (ChromoTek) essentially as described by the manufacturer. Benzonase (Millipore) treatment was performed before the preparation of soluble cell lysates to release chromatin-associated proteins.

## Phosphoproteomic analysis of chromatin-associated proteins

Chromatin-enriched protein fractions were prepared from HEK293 cells transfected either with control siRNA ("RIF1$^+$" cells) or RIF1 siRNA ("RIF1$^-$" cells) as described above. The chromatin pellet was resuspended in 1% SDS/100 mM tetraethylammonium bromide (TEAB). Proteins were reduced by 10 mM tris(2-carboxy-ethyl)phosphine for 1 h at 55°C, followed by alkylation for 30 min at room temperature in the presence of 17 mM iodoac-etamide. Proteins were then precipitated by addition of six volumes of cold acetone. Protein pellets were separated by centrifugation and resuspended in 100 mM TEAB buffer. Proteins were digested by either (i) trypsin (Pierce) alone or (ii) trypsin followed by Asp-N endopeptidase (Thermo Scientific). After removing insoluble materials by centrifugation, the supernatants were used for TMTduplex isobaric mass tag (Thermo Scientific) labeling reactions as described by the manufacturer. Roughly equal masses of TMT-labeled peptides from RIF1$^+$ and RIF1$^-$ cells were mixed, and fractionated by a high pH reverse-phase HPLC [71] into 24 concatenated fractions per digestion. For each fraction, phosphopeptides were enriched by MagResyn Ti-IMAC beads (Resyn Bioscience) using KingFisher Flex Magnetic Particle Processor (Thermo) as described [72]. The peptide samples were analyzed using an Orbitrap Fusion mass spectrometer (Thermo Scientific). Identification and relative quantification of phospho-peptides were carried out using MaxQuant software (ver. 1.5.3.8). Relative enrichment of phosphorylation at each identified residue was calculated by dividing the reporter signal intensity obtained from RIF1$^-$ cells by that from RIF1$^+$ cells. Peptides that did not associate with the Ti-IMAC resin were analyzed in parallel, and were used for normalization of the phospho-enriched datasets. For example, in the phosphorylated peptide sample, MCM4-phosphoS23 from RIF1$^-$ cells was measured as 2.92 times that from RIF1$^+$ cells. In RIF1$^-$ cells, however, total MCM4 on chromatin is 0.52 times that in the RIF1$^+$ cells (based on measurements of 64 peptides). So within the population of MCM4 molecules that are chromatin-associated, MCM4-phosphoS23 must be 5.63-fold increased by RIF1 depletion. Protein sequence database used was Uniprot human reference proteome (release 2015_08). The protein sequences and amino acid numbering of MCM proteins presented in Table 1 are based on the canonical sequences of Uniprot entries, P49736 (MCM2_HUMAN), P25205 (MCM3_HUMAN),

P33991 (MCM4_HUMAN), P33992 (MCM5_HUMAN), Q14566 (MCM6_HUMAN), and P33993 (MCM7_HUMAN).

## Flow cytometry

For DNA content analysis by flow cytometry, cells were recovered by trypsinization, and fixed in 70% ethanol. Cells were spun down, resuspended in PBS containing 50 μg/ml propidium iodide, 50 μg/ml RNase A, and 0.1% Triton X-100, and incubated at room temperature for 1 h (protected from light). DNA content per cell was analyzed on a Becton Dickinson FACSCalibur using FL2-A. Cell cycle distribution was measured using FlowJo software. Doublet discrimination was performed by setting a gate on a 2-D plot of FSC-A and FCS-H values.

For EdU pulse-label experiments, cells were labeled for 15 min with 20 μM 5-ethynyl-2′-deoxyuridine (EdU). Cells were collected, fixed, permeabilized, and stained essentially as described [73] using Click-iT Plus EdU Alexa Fluor 647 Flow Cytometry Assay Kit (Molecular Probes). DNA was stained with 0.5 μg/ml DAPI instead of propidium iodide. Cells were analyzed using a Becton Dickinson LSRFortessa flow cytometer. Data were analyzed by FlowJo software. Doublet discrimination was performed by setting a gate on a 2-D plot of FSC-A and FCS-H values.

Chromatin-associated proteins were analyzed by flow cytometry essentially as described [41] with modifications. After extraction, cells were stored in 70% ethanol at −20°C. We included a blocking step with 4% non-fat dried milk before the immunostaining. For antibody staining, cells were suspended in PBS with 0.1% Igepal CA-630 to a density of $10^7$ cells/ml, and 0.1 ml of cell suspension (=$10^6$ cells) were used. Antibodies were diluted in PBS containing 3% BSA. All the washing steps were performed using PBS with 0.1% Igepal CA-630. After immunostaining, cells were resuspended in PBS containing 0.1% Igepal CA-630 and 0.5 μg/ml DAPI and were kept in the dark prior to analysis by Beckon Dickinson LSRFortessa. Data were analyzed by FlowJo software. Doublet discrimination was performed by setting a gate on a 2-D plot of FSC-A and FCS-H values. To assess the effect of MG-132 and tautomycetin, culture medium was replaced with fresh culture medium containing either DMSO, 20 μM MG-132 (Invivogen), or 5 μM tautomycetin (Tocris), and cultivation was continued for 4 h prior to collecting cells.

For analysis of GFP-ORC1 mCherry-PCNA HeLa cells, cells were extracted and fixed essentially as above, except that the fixation was only 10 min. After washing in PBS containing 0.1% Igepal CA-630, cells were immediately stained with DAPI and fluorescence from GFP and mCherry analyzed by Becton Dickinson LSRFortessa.

## List of antibodies used in this study

The following antibodies were used for Western blotting:
1. RIF1; Bethyl Laboratories; A300-568A; rabbit polyclonal
2. GFP; Abcam; ab290; rabbit polyclonal
3. GFP; ChromoTek; rat monoclonal [3H9]
4. MCM4; Abcam, ab4459; rabbit polyclonal
5. p-S40-MCM2; Abcam; ab133243; rabbit monoclonal [EPR4170 (2)]
6. p-S53-MCM2; Bethyl Laboratories; A300-756A; rabbit polyclonal
7. PP1α; Santa Cruz Biotechnology; sc-271762; mouse monoclonal [G-4]

8. PP1β; Santa Cruz Biotechnology; sc-373782; mouse monoclonal [C-5]
9. PP1γ; Santa Cruz Biotechnology; sc-6108; goat polyclonal
10. Tubulin; Santa Cruz Biotechnology; sc-53030; rat monoclonal [YOL1/34]

Following antibodies were used for flow cytometric analysis of chromatin-associated proteins at indicated dilutions:
1. Cdt1 antibody; Abcam ab202067, at 1/100 dilution
2. FLAG tag antibody [M2]; Sigma F-1804, at 1/300 dilution
3. MCM3 antibody; Santa Cruz sc-9850, at 1/300 dilution
4. ORC2 antibody; Abcam ab31930, at 1/200 dilution
5. Alexa Fluor 488-conjugated anti-mouse IgG (ab150109), at 1/2,000 dilution
6. Alexa Fluor 568-conjugated anti-mouse IgG (ab175700), at 1/2,000 dilution
7. Alexa Fluor 647-conjugated anti-goat IgG (ab150135), at 1/2,000 dilution
8. Alexa Fluor 647-conjugated anti-rabbit IgG (ab150063), at 1/2,000 dilution

For immunoprecipitation of endogenous RIF1 protein, Abcam ab70254 was used. For microscopic observation of Cdc6 protein, Abcam ab188423 was used.

## Cell viability assay using CellTiter-Glo

A total of 1,000–5,000 cells in 100 μl of medium were transferred to wells of white 96-well plates (Greiner) with the indicated concentration of XL413. After 3 days of incubation, 100 μl of CellTiter-Glo assay reagent (Promega) was added to each well. Luminosity of each well, which represents total ATP level derived from viable cells, was measured using GloMAX luminometer (Promega). Values were normalized to the values without XL413 addition.

## DNA fiber analysis

Exponentially growing cells were pulse-labeled with 50 μM CldU (Sigma C6891) for 20 min followed by 250 μM IdU (I7125) for 20 min. Labeled cells were trypsinized and resuspended in cold PBS at an approximate density of $2.5 \times 10^5$ cells/ml; 2 μl of the cell suspension was placed toward the end of a regular microscope slide, pre-cleaned with EtOH 70%. Cells were lysed on the slide by adding 10 μl of fresh pre-warmed (30°C) spreading buffer (200 mM Tris–HCl pH 7.4, 50 mM EDTA, 0.5% SDS). After 6 min of incubation, the slides were tilted at a 10–15° angle to allow the DNA suspension to run slowly and spread down the slide. Slides were air-dried for at least 20 min and fixed in cold (−20°C) methanol–acetic acid (3:1). Slides were stored at 4°C before the following steps. DNA was denatured by incubation of the slides in 2.5 M HCl at RT in a Coplin staining jar for 30 min. Slides were rinsed 3× to completely remove HCl and blocked by incubation for 1 h in 1% BSA, 0.1% Triton X-100, PBS. Blocking was followed by incubation with primary antibodies diluted in blocking solution for 1 h at RT in humidity chamber (anti-CldU, Abcam ab6326, 1:100; anti-IdU, BD 347580, 1:100; anti-ssDNA, Millipore MAB3034, 1:100). Slides were washed three times in PBS and incubated with the following secondary antibodies diluted 1:300 in blocking solution (anti-rat IgG Alexa Fluor

594, Molecular Probes A-11007; anti-mouse IgG1 Alexa Fluor 488, Molecular Probes A-21121; anti-mouse IgG2a Alexa Fluor 350, Molecular Probes A-21130). Slides were air-dried and mounted with Prolong (Invitrogen). DNA fibers were imaged under a Zeiss Axio Imager.M2 microscope equipped with Zeiss MRm digital camera, with a Zeiss Plan-Apochromat 63×/1.40 Oil objective lens. Images were analyzed using ImageJ. To measure interorigin distances, adjacent replication origins within the same fiber were identified, and then distance in kb was calculated using the conversion factor 1 μm = 2.59 kb [74].

**Expanded View** for this article is available online.

## Acknowledgements

We thank David Stead at the Aberdeen Proteomics Service for help in mass spectrometry interpretation, and Raif Yücel and his team at the University of Aberdeen Iain Fraser Cytometry Centre for assistance with flow cytometry. We thank Robert Alver and Julian Blow at University of Dundee for advice on the use of tautomycetin. Peter Cherepanov of the Francis Crick Institute gifted XL413. Daniel Durocher of Lunenfeld-Tanenbaum Research Institute gifted DNA constructs. Work by ADD and SH was supported by Cancer Research UK Grant A13356, Cancer Research UK Programme Award A19059, and BBSRC grant (BB/K006304/1). AIL was supported by Wellcome Trust Awards (108058/Z/15/Z & 105024/Z/14/Z). This work was also supported by JSPS KAKENHI Grant # 16H04739, 25116004 to CO and 16J04327 to YO.

## Author contributions

S-iH, TL, JG, ZH, CO, SJB, AIL, and ADD conceived and designed the experiments. S-iH, JG, TL, ZH, Y-nO, and AE performed experiments. SH, JG, and TL analyzed the data. SH and ADD wrote the manuscript.

## Conflict of interest

The authors declare that they have no conflict of interest.

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
