## [Review Process File · EMBO Reports]

Manuscript EMBO-2016-41983

Human RIF1 and Protein Phosphatase 1 stimulate DNA replication origin licensing but suppress origin activation

Shin-ichiro Hiraga, Tony Ly, Javier Garzón, Zuzana Hořejší, Yoshi-nobu Ohkubo, Akinori Endo, Chikashi Obuse, Simon J. Boulton, Angus I. Lamond, and Anne D. Donaldson

Corresponding authors: Shin-ichiro Hiraga and Anne D. Donaldson, University of Aberdeen

Review timeline:

Submission Date:	28 July 2016
Editorial Decision:	25 August 2016
Revision Received:	08 November 2016
Accepted:	05 December 2016

Editor: Esther Schnapp

Transaction Report:

1st Editorial Decision

25 August 2016

Thank you for the submission of your manuscript to our journal. We have now received the full set of referee reports that is copied below as well as referee cross-comments.

As you will see, all referees acknowledge that the findings are interesting. They only raise a limited number of concerns, and I think that all of them should be addressed, except the major concerns of referee 3 as both referees 1 and 2 indicate in their cross-comments that this is not essential.

We would thus like to invite you to revise your manuscript with the understanding that the referee concerns must be fully addressed and their suggestions taken on board as outlined above. Please address all referee concerns in a complete point-by-point response. Acceptance of the manuscript will depend on a positive outcome of a second round of review. It is EMBO reports policy to allow a single round of revision only and acceptance or rejection of the manuscript will therefore depend on the completeness of your responses included in the next, final version of the manuscript.

REFeree REPORTS

Referee #1:

The authors set out to investigate the molecular mechanism allowing for human RIF1's control of DNA replication. Evidence in yeast suggests RIF1 controls replication through targeting of PP1 phosphatase to antagonize the DDK phosphorylation of the MCM helicase required for replication origin activation making the authors investigation of this interaction in a mammalian system very logical. The author's construction of RIF1 cDNAs with mutations in motifs known to interact with PP1 is key to this manuscript. Manipulation of RIF1 expression using a combination of siRNA

knockdown and exogenous overexpression of GFP-tagged RIF1 cDNA and mutant GFP-tagged RIF1 cDNA allows the authors to gain a transient look at RIF1 interactions in a human cell line. The authors show that the exogenously expressed GFP-RIF1 cDNA in their system does interact with all three isoforms of human PP1 and the mutations in the PP1 binding motifs abolish these PP1 interactions. The authors demonstrate that, like in yeast, human RIF1 and its interaction with PP1 is required for prevention of hyper-phosphorylation of MCM4 and MCM2. The authors dive into the biochemistry of the RIF1-PP1 interaction and identify the key residues of MCM4 and MCM2 that RIF1 targeted PP1 acts upon. The authors continue by observing minor defects in DNA replication in RIF1 overexpressing cells and ascribe this observation to reduced helicase activation, yet overexpression of the RIF1-PP1 interaction mutant does not curb this observation as their hypothesis predicts. The authors show RIF1 overexpression exacerbates the replication defects observed using a DDK inhibitor, which requires RIF1-PP1 interaction. The author's main finding results from the observation that RIF1 knockdown reduces EdU incorporation, when ablation of a predicted replication inhibitor would likely increase incorporation. The authors show that ORC1 and MCM3 are destabilized on chromatin in G1 when RIF1 is depleted, suggesting a role for human RIF1 in pre-RC preservation. The authors show that ORC1 and MCM3 chromatin stabilization is dependent on PP1 and proteasome action. The authors then hypothesize that inter-origin distances will be affected in RIF1 knockout cells and observe a slight increase by DNA fiber analysis.

Overall the authors fill two gaps in the literature with this paper: 1. They confirm in humans much of which has been observed in yeast, which lends to RIF1's conserved and important role in genome regulation. 2. They observe a new role for RIF1 in origin stabilization, which adds another chapter to the ever-growing RIF1 tome. I believe the intellectual merit of this paper is high and it should be accepted for publication, yet there are issues that will make the paper stronger, as outlined below.

- The cell line expressing mutant RIF1 defective for PP1 interaction is a very strong tool and is utilized well in the first part of the paper, but is forgotten in the second half. Overexpression of mutant RIF1 and knockdown of endogenous RIF1 would be very useful in showing RIF1-PP1 interactions are required for origin stabilization as much of the data provided relies on broad-spectrum PP1 inhibitors. These data would significantly strengthen the novel findings in this paper.
- Figure 7 - Actual pictures of the DNA fiber analysis would be helpful to interpret the extent to which RIF1 is affecting inter-origin distances. We also need to see the average size of all fibers for all experiments as that can strongly bias IOD calculations.
- Figure 1C - IP of endogenous RIF1 would provide a more convincing figure. PP1 isoform interactions seem very weak in the GFP-RIF1 cDNA, is this interaction similar to endogenous RIF1?
- The discussion tries to connect RIF1's control of replication timing to origin stabilization and possibly allow for more MCM complexes loaded at each origin, thus making these origins fire earlier, yet RIF1 binds mainly to late replicating domains according to ChIP data in Foti et al., (2016). The provided model could use revision to address how RIF1 might act at different chromatin domains to influence replication timing.

Minor Comments:

Table 1 – any way of knowing which PP1 isoform is acting at which residues?
Figure 4 - Why MCM3? 2 and 4 focused on previously

Referee #2:

This very clearly written paper studies the role of human RIF1 in the regulation of DNA replication. The question is important and appropriate for EMBO Reports and the experimented use state-of-the-art analyses to probe the regulation of DNA replication by RIF1. The main findings build on previous observations suggesting that RIF1 regulates replication initiation by interacting with PP1 phosphatase. The current submission demonstrates that the RIF1 protein interacts with all 3 isoforms of PP1 and recruits the phosphatase to pre-replication complexes. On chromatin, PP1 counters the phosphorylation of MCM, the replicative helicase, and ORC1, a member of the ORC complex required for preparing replication origins for initiation of DNA replication. Limiting MCM helicase phosphorylation prevents premature activation of the helicase and delays DNA replication. Limiting ORC1 phosphorylation prevents premature degradation of ORC1 and increases the frequency of

licensed origins. Hence, depletion of RIF1, in concert with inhibition of PP1, concomitantly deregulates the replication timing program and limits the abundance of activated replication origins. These observations provide a mechanistic basis for the reported roles of RIF1 in regulating replication timing and affecting genomic stability. I find the paper very interesting as it reports a significant advance of potential importance.

Several technical issues need to be addressed, as listed below:

1 / Figure 2B, lane 4: cells contain no RIF1 and were not exposed to DOX. Why do these cells contain a reduced level of S40 phosphorylated MCM2? Shouldn't these cells and the cells analyzed in lanes 2 and 6 phosphorylate S40 with a similar efficiency? Is the reduction reproducible?

2 / Figure 3C: do all cells treated with XL413 without siRIF1 incorporate EdU, or is the baseline of the "+GFP only" curve skewed to the right?

3 / Figure 6: depletion of RIF1 is reported to cause "statistically significant" increase in the median inter-origin distance. How significant was the change - can a p-value be calculated? Was the change accompanied by a change in replication fork progression as reported in other instances of reduced origin frequency, or was there a different effect due to the concomitant effects on replication timing? This point should at least be discussed.

Minor issues:

1/ It seems from the Methodology section that all the experiments reported in the main text were performed in the engineered Flp-in 293 system, but most figure legends do not state the source of the cells explicitly.

2/ Why were MEFs used in figure S5A, and what is the significance of their relative resistance compared to 293 cells? Is this a species difference or an effect of tissue culture adaptations? If the authors consider this an important observation, it might be good to discuss further.

Referee #3:

In this manuscript Hiraga et al investigate how human Rif1 affects DNA replication. In the first section they find that the PP1 dependent mechanism of Rif1 control is conserved between yeast and human and showed that all of the PP1 isoforms interact with Rif1 and consistently also displayed overlap in their activities. Then the authors go on and investigate the consequences of Rif1 overexpression and find synergistic effects with XL413 mediated DDK inhibition on DNA synthesis, consistent with the idea that Rif1 and DDK acts in the same pathway and work via PP1. These observations are important, as they clearly define the regulatory principles that control this system. In the second part the authors investigate the curious observation that RNAi mediated knockdown of Rif1 has a significant impact on the Orc1 phosphorylation levels, Orc1 stability and MCM2-7 loading. The authors observed that degradation of phosphorylated Orc1 through the known proteasome pathway is hyperactive in the absence of Rif1, as dephosphorylating of Orc1 is reduced. Crucially, Orc1 acts not only in replication, but also for heterochromatin formation and in regulating cilia formation, thus the implications are manifold. Moreover, reduced loading of the replicative helicase core complex MCM2-7 is known to affect DNA synthesis only mildly, as each cell cycle an excess of MCM2-7 becomes loaded on DNA, which serves as dormant origins. These origins have important functions in case of replication fork blockage and allow in this case reestablishment of replication forks. In the absence of dormant origins cells become hypersensitive to DNA damage, as they have no way to reload their helicase during ongoing S-phase. In particular stem cells are hypersensitive to dormant origin interference. Understanding the circuits that regulate loading of MCM2-7 have therefore major importance for a large number of scientists and clinicians. However, the levels of MCM2-7 loading have also been suggested to affect the timing of replication (Nick Rhind), therefore this is an important knowledge, which needs to be considered when interpreting the manifold studies that study the role of Rif1 in replication timing. To enhance this 2nd part of the work, and a few other sections, I would suggest the following major and minor modifications.

Major concerns:

A key finding is that Orc1 becomes protected from degradation through Rif1 and PP1 mediated de-phosphorylation. However, the mechanism is not clear. How are Rif1/PP1 being recruited to Orc1? What amino acids in Orc1 become phosphorylated upon Rif1 knockdown? Does mutating these sites have an influence on MCM2-7 loading or cell cycle progression? Answering at least some of these questions would enhance the manuscript.

Minor points:

- Figure 1B. It would be easier to judge the nuclear localisation if the DAPI image is shown as well.
- Figure 1C, The Rif1 IP shows weak bands for PP1 a,b,c, but to judge the background binding towards Rif1-pp10bs a longer exposure would be useful.
- The quality of the work is in general very high, but I wonder if a single siRNA is sufficient to exclude off target effects.
- Figure 2B. The authors have no loading control. Having this, or a ponceau stain would make the comparison easier.
- Figure 4. Could the authors show what happens to Cdc6, geminin and Cdt1 - as these factors also regulate MCM2-7 loading
- Figure 4D. Some of the effects are less pronounced. Is this due to transfection efficiencies? Could this be discussed?
- Figure 6 or supplementary figures. I wonder if it would be useful to test a panel of DNA damaging drugs in the cells used here to showcase the effect of Rif1 depletion on cell growth, DNA content and DNA synthesis rates using DNA combing.
- Figure S2 and S3A appear to miss the gel sections. S3B shows the identical figure as in Fig.2C. I would recommend to describe this clearly in the figure legend.

1st Revision - authors' response

08 November 2016

We thank the Reviewers for their interest in the work and their useful comments, which we address below (in italics):

Referee #1:

The authors set out to investigate the molecular mechanism allowing for human RIF1's control of DNA replication. Evidence in yeast suggests RIF1 controls replication through targeting of PP1 phosphatase to antagonize the DDK phosphorylation of the MCM helicase required for replication origin activation making the authors investigation of this interaction in a mammalian system very logical. The author's construction of RIF1 cDNAs with mutations in motifs known to interact with PP1 is key to this manuscript. Manipulation of RIF1 expression using a combination of siRNA knockdown and exogenous overexpression of GFP-tagged RIF1 cDNA and mutant GFP-tagged RIF1 cDNA allows the authors to gain a transient look at RIF1 interactions in a human cell line. The authors show that the exogenously expressed GFP-RIF1 cDNA in their system does interact with all three isoforms of human PP1 and the mutations in the PP1 binding motifs abolish these PP1 interactions. The authors demonstrate that, like in yeast, human RIF1 and its interaction with PP1 is required for prevention of hyper-phosphorylation of MCM4 and MCM2. The authors dive into the biochemistry of the RIF1-PP1 interaction and identify the key residues of MCM4 and MCM2 that RIF1 targeted PP1 acts upon. The authors continue by observing minor defects in DNA replication in RIF1 overexpressing cells and ascribe this observation to reduced helicase activation, yet overexpression of the RIF1-PP1 interaction mutant does not curb this observation as their hypothesis predicts. The authors show RIF1 overexpression exacerbates the replication defects observed using a DDK inhibitor, which requires RIF1-PP1 interaction. The author's main finding results from the observation that RIF1 knockdown reduces EdU incorporation, when ablation of a predicted replication inhibitor would likely increase incorporation. The authors show that ORC1 and MCM3 are destabilized on chromatin in G1 when RIF1 is depleted, suggesting a role for human RIF1 in pre-RC preservation. The authors show that ORC1 and MCM3 chromatin stabilization is dependent on PP1 and proteasome action. The authors then hypothesize that inter-origin distances will be affected in RIF1 knockout cells and observe a slight increase by DNA fiber analysis.

Overall, the authors fill two gaps in the literature with this paper: 1. They confirm in humans much of which has been observed in yeast, which lends to RIF1's conserved and important role in genome regulation. 2. They observe a new role for RIF1 in origin stabilization, which adds another chapter to the ever-growing RIF1 tome. I believe the intellectual merit of this paper is high and it should be accepted for publication, yet there are issues that will make the paper stronger, as outlined below.

- The cell line-expressing mutant RIF1 defective for PP1 interaction is a very strong tool and is utilized well in the first part of the paper, but is forgotten in the second half. Overexpression of mutant RIF1 and knockdown of endogenous RIF1 would be very useful in showing RIF1-PP1 interactions are required for origin stabilization as much of the data provided relies on broad-spectrum PP1 inhibitors. These data would significantly strengthen the novel findings in this paper.

This is a very good suggestion. We have added a panel ('Extended View' Fig. EV3A in the revised version) quantifying the effect on licensing of RIF1 depletion with or without induction of GFP-RIF1 or RIF-pp1bs, assessed by measuring chromatin-associated MCM4 in western blot experiments. We find that RIF1 depletion mildly but reproducibly impairs licensing, and that this effect is rescued by induction of GFP-RIF1 but not GFP-RIF1pp1bs. This result is as predicted by our model, in which RIF1 promotes origin licensing by directing PP1 to dephosphorylate and stabilize ORC1. The effect is small when looking by western blotting at all cells in the population, and we would have liked to examine the effects in relation to cell cycle stage. We did attempt this experiment by examining the effect of RIF1 depletion with GFP-RIF1/GFP-RIF1pp1bs induction using flow cytometry of MCM3 in permeabilised DAPI-stained cells. Unfortunately however, we have been unable to obtain satisfactory results when using the cell cycle analysis-flow cytometry method with our induced GFP-RIF1 system for this experiment. (Probably for reasons to do with heterogeneity of expression level of the induced protein, we found that a high degree of scatter makes the results of such experiments uninterpretable.)

- Figure 7 - Actual pictures of the DNA fiber analysis would be helpful to interpret the extent to which RIF1 is affecting inter-origin distances. We also need to see the average size of all fibers for all experiments as that can strongly bias IOD calculations.

We have completely revised this Figure (now Fig. 6), and now present a new, more complete set of data, examining the effect on IOD both when replication is unimpeded and in the presence of Hydroxyurea to examine the effect of RIF1 depletion on dormant origins. We also now include specimen pictures of our fiber analyses. The substantially revised text describing this Figure is on pages 13-14 of the Revised Manuscript. The histogram now shown in Fig. 6B provides a much more detailed view of the distribution of inter-origin distances. We are not clear however exactly what the Reviewer means by 'average size of all fibers'. In these experiments the fibers examined are generally a few hundred kb in length, but the precise end-to-end length of each fiber being analysed is not a number easily obtained in such fiber methods. While the fiber size will indeed affect measurements (since you obviously can't detect an IOD greater than the fibre size), there's no reason to expect a different average fibre size in control and RIF1-depleted cells, so the differences we see are still valid. We now discuss on page 15 the implications for our results of the fact that fiber analyses can assess local effects on replication initiation, but not effects on entire origin clusters or chromosome domains.

- Figure 1C - IP of endogenous RIF1 would provide a more convincing figure. PP1 isoform interactions seem very weak in the GFP-RIF1 cDNA, is this interaction similar to endogenous RIF1?

We can IP endogenous RIF1, but despite several attempts have been unable to detect PP1 in the IP samples, as now explained towards the bottom of page 5 of the revised manuscript. We suspect that this result reflects the fact that many different PP1-interacting proteins compete for a limited pool of human PP1 proteins, so that RIF1 overexpression is needed for the amount of PP1 recovered in IP samples to be above our detection threshold. An alternative possibility is that the RIF1 epitope recognized by the antibody we used for IP is masked in the fraction of RIF1 that interacts with PP1 (and indeed, we were only able to pull down around 50% of endogenous RIF1, with a significant proportion remaining in the supernatant despite an excess of antibody). As now covered towards the bottom of page 5, several large-scale analyses of the PP1 interactome already identified endogenous RIF1 as binding to PP1 under physiological conditions, confirming that the interaction

we observe in Fig. 1 C does reflect a genuine biological interaction. For example, one investigation from co-author Angus Lamond's lab identified RIF1 as associating with PP1 that was isolated using microcystin-coupled beads (Moorhead et al, 2008). We now mention these earlier studies to clarify the situation (bottom of Page 5). Taken together with our GFP-RIF1 pull-down experiment presented in this manuscript, we believe that RIF1-PP1 interaction has been well demonstrated.

- The discussion tries to connect RIF1s control of replication timing to origin stabilization and possibly allow for more MCM complexes loaded at each origin, thus making these origins fire earlier, yet RIF1 binds mainly to late replicating domains according to ChIP data in Foti et al., (2016). The provided model could use revision to address how RIF1 might act at different chromatin domains to influence replication timing.

The Discussion has been completely rewritten and now covers the relationship to previous replication timing studies more explicitly. We suggest a specific model for how RIF1 might affect both replication licensing and replication timing, on pages 16-17.

Minor Comments:

Table 1 – any way of knowing which PP1 isoform is acting at which residues?

There's no way to know this from our existing data. To address this issue properly it would be necessary to carry out proteomic analysis on cells depleted for the individual siPP1 isoforms, which would represent a considerable undertaking beyond the scope of this manuscript. (And in practice, the likelihood that isoforms can substitute or compensate for one another, see Appendix Fig. S1, means that to draw clear conclusions it would probably be necessary to carry out proteomic analyses in two-way compared to three-way depleted cells, to enable analysis of effects when just a single PP1 isoform is present.)

Figure 4 - Why MCM3? 2 and 4 focused on previously

Based on the proteomic results summarized in Fig. 4A we presume that that all six MCMs behave similarly in terms of loading at origins. We focused in MCM3 in the flow cytometry analyses for technical reasons: First, flow cytometry analysis of MCM3 chromatin association was shown to work well, using the antibody we have utilised, in the paper that established the method (Ref 41: Haland et al 2015). Second, since MCM4 is strongly phosphorylated, we were concerned by the possibility that altered phosphorylation level might affect antibody detection and therefore quantification using the flow cytometry approach.

Referee #2:

This very clearly written paper studies the role of human RIF1 in the regulation of DNA replication. The question is important and appropriate for EMBO Reports and the experimented use state-of-the-art analyses to probe the regulation of DNA replication by RIF1. The main findings build on previous observations suggesting that RIF1 regulates replication initiation by interacting with PP1 phosphatase. The current submission demonstrates that the RIF1 protein interacts with all 3 isoforms of PP1 and recruits the phosphatase to pre-replication complexes. On chromatin, PP1 counters the phosphorylation of MCM, the replicative helicase, and ORC1, a member of the ORC complex required for preparing replication origins for initiation of DNA replication. Limiting MCM helicase phosphorylation prevents premature activation of the helicase and delays DNA replication. Limiting ORC1 phosphorylation prevents premature degradation of ORC1 and increases the frequency of licensed origins. Hence, depletion of RIF1, in concert with inhibition of PP1, concomitantly deregulates the replication timing program and limits the abundance of activated replication origins. These observations provide a mechanistic basis for the reported roles of RIF1 in regulating replication timing and affecting genomic stability.

I find the paper very interesting as it reports a significant advance of potential importance. Several technical issues need to be addressed, as listed below:

1/ Figure 2B, lane 4: cells contain no RIF1 and were not exposed to DOX. Why do these cells contain a reduced level of S40 phosphorylated MCM2? Shouldn't these cells and the cells analyzed in lanes 2 and 6 phosphorylate S40 with a similar efficiency? ? Is the reduction reproducible?

This effect on MCM2 Serine 40 phosphorylation is not reproducible, as now shown in a biological repeat of this experiment presented as Fig. EV1 (and mentioned in the text on page 6).

2 / Figure 3C: do all cells treated with XL413 without siRIF1 incorporate EdU, or is the baseline of the "+GFP only" curve skewed to the right?

Comparing this histogram in Fig. 3C with the 2-dimensional plots of the same data shown in Fig. EV2B (panel i), it becomes clear that this apparent 'skew' is caused by cells struggling to finish (not start) S phase. The fact that DDK inhibition leads to a buildup of cells in late S phase has been previously described in ref 40 (Koltun et al 2012). Interestingly, RIF1 removal partially suppresses this effect also, as is clear from inspection of Fig. EV2B panel ii, further confirming that DDK and RIF1 act in opposition in S phase control. We now explicitly mention this aspect of the data, close to the top of page 9.

3 / Figure 6: depletion of RIF1 is reported to cause "statistically significant" increase in the median inter-origin distance. How significant was the change - can a p-value be calculated?

We now include this p-value in the revised text on page 13 ($p = 0.001$).

Was the change accompanied by a change in replication fork progression as reported in other instances of reduced origin frequency, or was there a different effect due to the concomitant effects on replication timing? This point should at least be discussed.

We have measured fork speed in our fiber experiments and find that siRIF1 does not have a significant effect, as now presented in Fig. EV4B. As explained in our response to Reviewer 1, we have now provided a more complete and detailed dataset analyzing effects on origin frequency (in revised Fig. 6). In the revised Discussion we also now provide a more detailed Discussion of the relationship of changes in IOD to other effects of RIF1, in particular effects on EdU incorporation (bottom half of page 15) and on replication timing (page 16).

Minor issues:

1/ It seems from the Methodology section that all the experiments reported in the main text were performed in the engineered Flp-in 293 system, but most figure legends do not state the source of the cells explicitly.

We now state explicitly the cell line being used, in each legend.

2/ Why were MEFs used in figure S5A, and what is the significance of their relative resistance compared to 293 cells? Is this a species difference or an effect of tissue culture adaptations? If the authors consider this an important observation, it might be good to discuss further.

We have removed both of the Supplementary Figures that showed analyses of MEF cells, as they did not make any major points and indeed raised questions about how the results related to those in the rest of the paper, which describe effects in human cell lines.

Referee #3:

In this manuscript Hiraga et al investigate how human RIF1 affects DNA replication. In the first section they find that the PP1 dependent mechanism of RIF1 control is conserved between yeast and human and showed that all of the PP1 isoforms interact with RIF1 and consistently also displayed overlap in their activities. Then the authors go on and investigate the consequences of RIF1 overexpression and find synergistic effects with XL413 mediated DDK inhibition on DNA synthesis, consistent with the idea that RIF1 and DDK acts in the same pathway and work via PP1. These observations are important, as they clearly define the regulatory principles that control this

system. In the second part the authors investigate the curious observation that RNAi mediated knockdown of RIF1 has a significant impact on the ORC1 phosphorylation levels, ORC1 stability and MCM2-7 loading. The authors observed that degradation of phosphorylated ORC1 through the known proteasome pathway is hyperactive in the absence of RIF1, as dephosphorylation of ORC1 is reduced. Crucially, ORC1 acts not only in replication, but also for heterochromatin formation and in regulating cilia formation, thus the implications are manifold. Moreover, reduced loading of the replicative helicase core complex MCM2-7 is known to affect DNA synthesis only mildly, as each cell cycle an excess of MCM2-7 becomes loaded on DNA, which serves as dormant origins. These origins have important functions in case of replication fork blockage and allow in this case reestablishment of replication forks. In the absence of dormant origins cells become hypersensitive to DNA damage, as they have no way to reload their helicase during ongoing S-phase. In particular stem cells are hypersensitive to dormant origin interference. Understanding the circuits that regulate loading of MCM2-7 have therefore major importance for a large number of scientists and clinicians. However, the levels of MCM2-7 loading have also been suggested to affect the timing of replication (Nick Rhind), therefore this is an important knowledge, which needs to be considered when interpreting the manifold studies that study the role of RIF1 in replication timing.

We agree that the effects on MCM loading might well impact on replication timing, and on page 17 of the Discussion (second paragraph) we now cover the results from the Rhind lab concerning the relationship between RIF1 depletion, replication timing and MCM loading.

To enhance this 2nd part of the work, and a few other sections, I would suggest the following major and minor modifications.

Major concerns:

A key finding is that ORC1 becomes protected from degradation through RIF1 and PP1 mediated dephosphorylation. However, the mechanism is not clear. How are RIF1/PP1 being recruited to ORC1? What amino acids in ORC1 become phosphorylated upon RIF1 knockdown?

Detailed dissection of the mechanism is beyond the scope of this study, especially as the phosphorylation sites that direct ORC1 destruction have not yet been identified. We do now mention in the Discussion the interesting proximity of ORC1 Ser273, one of the phosphorylation sites that is substantially affected by RIF1, to a consensus Destruction box and a PP1 interaction motif. We outline the potential significance of the juxtaposition of these three features on pages 15-16.

Does mutating these sites have an influence on MCM2-7 loading or cell cycle progression?

This is a very interesting question, but setting up ORC1 CRISPR-based mutagenesis or an ORC1 replacement system would require the establishment and characterization of a new set of cell lines, which would take many months and is beyond the scope of this study.

Answering at least some of these questions would enhance the manuscript.

Minor points:

Figure 1B. It would be easier to judge the nuclear localisation if the DAPI image is shown as well.

We have replaced all the panels in Fig. 1B and now include a DAPI image.

Figure 1C, The RIF1 IP shows weak bands for PP1 a,b,c, but to judge the background binding towards RIF1-pp10bs a longer exposure would be useful.

The detection system employed here was a cooled-CCD camera-based system with ECL substrate. Although it is a high sensitivity system, obtaining data as in Fig. 1C almost hits its detection limit. The Figure shows a 15 min exposure, and extending the exposure time further is not practical due to background noise caused by environmental heat. As these conditions produce no visible bands or background signal in the RIF1pp1bs pull-down lanes, we conclude that there's no background binding at any detectable level. As explained in our reply to Reviewer 1 and discussed on page 5 of

the manuscript, we suspect that the low level of PP1 reflects the fact that many different PP1-interacting proteins compete for actually quite a limited pool of human PP1 proteins.

The quality of the work is in general very high, but I wonder if a single siRNA is sufficient to exclude off target effects.

The synonymous mutations in the siRNA-resistant RIF1 construct mean that we have no flexibility in design of siRIF1 RNAs. However, the fact that the GFP-RIF1 complements the phenotypes tested (Fig. 2A (ii), Fig. EV3A) implies that observations are not due to off-target effects.

Figure 2B. The authors have no loading control. Having this, or a ponceau stain would make the comparison easier.

Indeed in the original manuscript we had overlooked to explain how we ensured equivalent loading in all lanes, which is now explained on pages 20-21. We also now include a protein loading control panel in the (new) Fig. EV1, which shows a biological repeat of the experiment of Fig. 2B.

Figure 4. Could the authors show what happens to Cdc6, geminin and Cdt1 - as these factors also regulate MCM2-7 loading.

We have tested for effects of RIF1 depletion on the cell cycle control mechanisms operating on Cdc6 and Cdt1. We present the results in Fig. EV5, and close to the top of page 16 we explain that RIF1 does not seem to have a major effect on either protein. Although geminin has been reported to be phosphorylated, we are not aware of any evidence that geminin's activity in replication control is regulated by phosphorylation, and so have no clear prediction to test about how geminin might be affected by RIF1-PP1.

Figure 4D. Some of the effects are less pronounced. Is this due to transfection efficiencies? Could this be discussed?

RIF1 depletion does reproducibly affect MCM loading (Fig. 4C & Fig. EV3E) to a greater extent than it does ORC1 loading (Fig. 4D). Given that one ORC complex is believed to load multiple MCMs, it is as expected that MCM loading is generally more seriously affected than ORC1 loading, as now explained on page 15 lines 1-3. Differing transfection efficiencies between the HEK293 and HeLa-based cell lines may indeed cause variation between experiments, as now mentioned on page 12 (lines 8-9, in the Discussion of Fig.4.)

Figure 6 / supplementary figures: I wonder if it would be useful to test a panel of DNA damaging drugs in the cells used here to showcase the effect of RIF1 depletion on cell growth, DNA content and DNA synthesis rates using DNA combing.

This is an interesting suggestion, but it is unclear what information DNA combing/fibre analysis would provide about cell growth, DNA content, or even overall DNA synthesis rate. It is certainly true that RIF1 is emerging as involved in the control of pathways affected by various DNA damaging drugs (-- including UFB resolution [Hengeveld et al Dev Cell 2015 34: 466 ref 65]; DSB repair [Chapman et al 2013 Mol. Cell 49: 858 ref 62; Zimmermann et al 2013 Science 339: 700 ref 64; Feng et al 2013 JBC 288: 11135 ref 64; Escribano-Diaz et al 2013 Mol Cell 49: 872 ref 34]; and fork protection [Chaudhuri et al Nature 2016 535: 382]). Partly because of the complex effects of RIF1 in these interrelated pathways, if it were to be done properly the screen being suggested here would represent a very substantial study, involving many assays other than simply combing. In fact to be useful this investigation would in effect form an entire separate study, involving dissection of multiple aspects of RIF1 biological function whose elucidation lies beyond the scope of this study.

Figure S2 and S3A appear to miss the gel sections. S3B shows the identical figure as in Fig.2C. I would recommend describing this clearly in the figure legend.

The Supplementary Figure panels mentioned are all taken from the main Figures, and are reproduced in the Supplementary (now 'Appendix') Figures simply to help clarify exactly what is being quantitated. Those in original Fig S2 (now Appendix Fig. S1) are from Fig. 2B (i) lower left

panels, while those in original Fig. S3 (now Appendix Fig. S2) are from the Fig. 2B upper (MCM4) panels. We now state this clearly in the Appendix Figure legends.

2nd Editorial Decision

5 December 2016

I am very pleased to accept your manuscript for publication in the next available issue of EMBO reports. Thank you for your contribution to our journal.

REFEREE REPORTS

Referee #1:

The authors have now addressed all of my criticisms satisfactorily. I look forward to the opportunity to cite this interesting work presenting a novel role for Rif1 in regulating DNA replication.

Referee #2:

The revision has addressed all my concerns and the main concerns of the other reviewer, I recommend publication in the current form.

Corresponding Author Name: Shin-ichiro Hiraga and Anne D. Donaldson

Manuscript Number: EMBOR-2016-41983V3